# Impact of Maternal Exposure to SARS-CoV-2 on Immunological Components of Breast Milk

**DOI:** 10.3390/ijms26062600

**Published:** 2025-03-13

**Authors:** Nayara Gomes Graciliano, Marília Oliveira Fonseca Goulart, Alane Cabral Menezes de Oliveira

**Affiliations:** 1Institute of Biological and Health Sciences, Federal University of Alagoas, Maceió 57072-900, Alagoas, Brazil; 2Institute of Chemistry and Biotechnology, Federal University of Alagoas, Maceió 57072-900, Alagoas, Brazil; 3College of Nutrition, Federal University of Alagoas, Maceió 57072-900, Alagoas, Brazil

**Keywords:** COVID-19, milk human, immunoglobulins, cytokines, cells

## Abstract

COVID-19, caused by SARS-CoV-2, has become a global public health threat. Although no replication-competent virus has been found in breast milk samples, breastfeeding practices during the pandemic were impacted. It is well known that breast milk is adapted to meet the needs of infants, providing the appropriate amounts of nutrients and various bioactive compounds that contribute to the maturation of the immune system and antioxidant protection, safeguarding infants against diseases. While its composition is variable, breast milk contains immune cells, antibodies, and cytokines, which have anti-inflammatory, pro-inflammatory, antiviral, and antibacterial properties that strengthen infant immunity. Since COVID-19 vaccines have not yet been approved for infants under six months of age, newborns rely on the passive transfer of antibodies via the placenta and breast milk to protect them against severe SARS-CoV-2 infection. Several studies that analyzed breast milk samples in the context of COVID-19 have demonstrated that a strong antibody response is induced following maternal infection with SARS-CoV-2. Therefore, this review aims to provide a comprehensive overview of the impact of maternal exposure to SARS-CoV-2 through natural infection and/or vaccination on the immunological composition of breast milk based on the studies conducted on this topic.

## 1. Introduction

In December 2019, severe acute respiratory syndrome coronavirus 2 (SARS-CoV-2) was discovered in Wuhan, China. The outbreak of coronavirus disease 2019 (COVID-19) has become a global public health threat due to its rapid spread and the risk of complications and deaths associated with respiratory diseases. The World Health Organization (WHO) characterized it as a pandemic on 11 March 2020 [1,2].

COVID-19 is caused by SARS-CoV-2, a virus of the Coronaviridae family, which is transmitted through the inhalation of droplets released through the nose or mouth when an infected person (pre-symptomatic, asymptomatic, or symptomatic) breathes, coughs, sneezes, or speaks, through contact with fomites, or even through fecal–oral transmission [3,4,5]. The virus has a spherical morphology consisting of a nucleocapsid that protects the genetic material (single-stranded RNA) and an outer envelope [6]. Its genome encodes a series of structural proteins (membrane protein—M, nucleocapsid protein—N, envelope protein—E, and spike protein—S), non-structural proteins (most of which comprise the viral replication and transcription complex), and accessory proteins. The S protein forms trimers that project from the viral envelope, being the primary determinant of coronavirus tropism: its ability to infect a particular type of cell or tissue [4] (Figure 1).

Over time, numerous SARS-CoV-2 mutant variants have been identified, with the Variants of Concern (VOC)—Alpha (lineage B.1.1.7), Beta (lineage B.1.351), Gamma (lineage P.1), Delta (lineage B.1.617.2), and Omicron (lineage B.1.1.529)—standing out due to their increased transmissibility and/or virulence potential [5]. The primary concern regarding the emergence of new SARS-CoV-2 variants lies in the reduced effectiveness of vaccines and natural immunity, driven by genomic alterations, particularly in the coding regions of the S protein, which increase viral fitness compared to ancestral strains [7].

The disease caused by SARS-CoV-2 can manifest in different clinical forms, ranging from asymptomatic or mild cases to more severe cases that can progress to acute respiratory distress syndrome (ARDS), a hyper-inflammatory response, and widespread damage to multiple organs, potentially fatal [6,8]. Children and neonates commonly present mild cases of COVID-19. Still, some children may present severe and lethal forms of the disease, such as multisystem inflammatory syndrome in children (MIS-C) and neonates (MIS-N), which manifests as a post-infectious inflammatory condition associated with abnormal immune function, left ventricular cardiac dysfunction, coronary artery aneurysms, atrioventricular block, and clinical deterioration with the involvement of different organs [9].

The host response to SARS-CoV-2 infection involves innate and adaptive T and B cell immunity and the antiviral-neutralizing antibody response [10]. During infection, innate immune cells (macrophages, monocytes, neutrophils, dendritic cells, and innate lymphoid cells—ILCs, such as natural killer cells—NK cells) produce cytokines and chemokines, which limit virus replication and induce the death of infected cells [5,11]. The adaptive immune response is carried out by B cells, which produce immunoglobulins (IgA, IgM, and IgG) specific to viral antigens, such as anti-protein S immunoglobulins (targeting the S1 protein and RBD) and anti-protein N, in addition to T cells (CD4^+^ and CD8^+^), which mainly target the S, M, and N proteins and the non-structural protein (including NSP3 and NSP4) of SARS-CoV-2 [9,12].

T cell and B cell responses are more durable than antibodies, with CD4^+^ and CD8^+^ memory responses detected in approximately 90% and 70% of individuals, respectively. Thus, although reinfection is possible after recovery from COVID-19 and/or after vaccination against SARS-CoV-2, there is an immunological memory related to the different immunological compartments (circulating antibodies, memory B cells, SARS-CoV-2-specific CD4^+^ and CD8^+^ T cells) that can last 8 months or more [13].

Although breastfeeding practices were severely impacted by COVID-19, particularly early in the pandemic [14], more recent studies show that COVID-19 diagnosis did not affect breastfeeding rates in high- and upper-middle-income countries [15,16]. Giuliani et al. explored the association between breastfeeding breast milk and neonatal COVID-19 test positivity and the risk of SARS-CoV-2 transmission through breastfeeding compared to feeding expressed human milk in a multinational cohort. Their study showed that breastfeeding in breastfeeding mothers with a confirmed diagnosis of COVID-19 and immediate skin-to-skin contact was not associated with an increased risk of neonatal test positivity in settings where mothers wore masks and washed their hands before touching their newborns [17]. A systematic review evaluating the association between breastfeeding practice and infants who tested positive for SARS-CoV-2 infection identified no association [18].

For more than 20 years, the protective power of breast milk against many diseases, particularly against viral infections, has been known due to the richness of its constituents related to the immune system [19]. Breast milk protects against infections through multiple mechanisms, including strengthening the infant’s epithelium barrier via growth factors; transferring antimicrobial agents like lactoferrin, lysozyme, and oligosaccharides; promoting beneficial microbial growth; and conveying maternal antigen-specific immunity through lymphocytes and antibodies [20]. It is documented that the composition of breast milk dynamically adjusts to infections, enhancing the infant’s immune defenses [21].

Studies have shown that maternal and/or infant infections rapidly increase leukocyte levels in breast milk, which return to baseline after recovery [22,23]. Zheng et al. reported that infant respiratory infections promote the migration of anti-inflammatory macrophages into breast milk, suggesting an additional mechanism of infant protection [24]. Chemokines and their receptors play an essential role in innate immunity against viral infections [25,26]. Farquhar et al. found that higher concentrations of macrophage inflammatory protein-1β (MIP-1β) and stromal cell-derived factor-1α (SDF-1α) in breast milk were linked to a lower risk of vertical human immunodeficiency virus 1 (HIV-1) transmission, while elevated RANTES (regulated upon activation, expressed, and secreted by normal T cells) levels were associated with increased risk, regardless of viral RNA levels in breast milk [26].

Breast milk is considered the gold standard for infant nutrition, characterized as a complex biological fluid with a dynamic, highly modifiable nature. It is biologically active and species-specific, and although it is similar across all women, its composition constantly adapts to meet the infant’s specific needs [27]. The composition of breast milk varies according to the stage of lactation (colostrum, transitional milk, and mature milk) and between full-term and preterm newborns. Factors such as ethnicity, diet, maternal age, parity, maternal and child health status, environment, and milk management (collection, storage, and pasteurization) also influence its composition. Its regulation is achieved through a complex interplay of hormonal mechanisms and feedback signals from the infant, who actively contributes to adjusting milk composition through both mechanical and biochemical cues [28].

Among the various immunologically active components found in breast milk (Figure 2), which provide active and passive immunity to the infant, the following are particularly notable: immunoglobulins IgA and their secretory form SIgA, IgM, and IgG, cytokines and chemokines, growth factors and various cellular components, such as leukocytes, macrophages, dendritic cells, NK cells, ILCs, and B and T lymphocytes. These components play an essential role in the maturation and development of the infant’s immune system [29].

Immunoglobulins in breast milk provide the newborn with passive immunity acquired through maternal antibodies that reflect the mother’s antigenic repertoire throughout her life. This phenomenon is significant because newborns are not yet capable of producing antibodies and rely on IgG, transferred from maternal circulation through the placenta during pregnancy, as temporary protection against pathogens to which the mother has been exposed [30,31]. All immunoglobulins (IgA, IgG, IgM, IgD, and IgE) have been found in breast milk. However, IgA/SIgA is the most abundant immunoglobulin in breast milk at any stage of lactation [32]. Figure 3 describes the main functions of breast milk immunoglobulins.

Cytokines are a group of small bioactive proteins, glycoproteins, and peptides that can be classified into several categories, including chemokines, interferons (IFNs), interleukins (ILs), transforming growth factors (TGFs), colony-stimulating factors (CSFs), tumor necrosis factors (TNFs), and adipokines. These molecules are essential for cellular communication and function within a network to regulate various biological processes, including innate and acquired immunity, hematopoiesis, inflammation and repair, and cell proliferation, primarily through extracellular signaling. Cytokines are mainly produced by immune system cells and are present in breast milk, where they are crucial for the development of the infant’s immune system [33]. Additionally, growth factors found in breast milk, such as vascular endothelial growth factor (VEGF), hepatic growth factor (HGF), and epidermal growth factor (EGF), play key roles in various biological functions, promoting the growth and protection of the infant’s gastrointestinal tract [34].

Advanced techniques, such as multiparametric flow cytometry, have significantly improved our understanding of the cellular composition of breast milk [35]. Breast milk is known to contain a heterogeneous mix of cells derived from both the breast and maternal blood, including leukocytes, mammary glandular epithelial cells, and stem cells [36], as well as bacteria [37], fungi, and yeasts [38]. Cells transferred to infants through breastfeeding help program the newborn’s immune response and promote the colonization of beneficial intestinal bacteria, thereby protecting against infections and inflammation in the early stages of life while also contributing to the maturation of the infant’s immune system [39].

Although the mechanism involved in the transfer of immune cells from breast milk to the circulatory system of the human newborn remains unclear, studies using animal models provide strong evidence supporting the effector role of immune cell trafficking from the mammary gland to the mucosal surfaces or tissues of the infant [40,41,42,43,44]. Animal studies have demonstrated the presence of immune cells originating from breast milk in an infant’s thymus, lymph nodes, spleen, Peyer’s patches, brain, and intestine [42]. The transfer of maternal cells into the neonatal circulation and their subsequent establishment in the infant’s organs is known as maternal microchimerism. This occurs during pregnancy through the transfer of maternal cells to the fetus via the placenta and later during breastfeeding [39].

Breastfeeding has been recognized as a key component of newborn health [45], as ingesting immunomodulatory compounds from breast milk protects against the incidence and severity of gastrointestinal and respiratory diseases through the passive transfer of maternal innate and adaptive immunity [46]. Since COVID-19 vaccines have not yet been approved for infants under 6 months of age, passive immunity conferred by the placenta and breast milk are the only ways to provide immunological defense against severe SARS-CoV-2 infection during the first 2 to 6 months of postnatal age [47,48].

Although SARS-CoV-2 infection has reached an endemic phase with periodic outbreaks, population immunity acquired through natural infection and/or vaccination declines over time, increasing the risk for vulnerable groups [49]. Additionally, vaccine booster uptake has been decreasing. As of March 2024, only 23% of adults in the United States reported receiving the updated 2023–2024 COVID-19 vaccine [50]. Between October 2023 and April 2024, approximately 70% of pregnant women had not received this vaccine before or during pregnancy [51]. In this context, achieving adequate vaccination coverage among pregnant and lactating women remains a global challenge [52].

Studies analyzing breast milk samples in the context of COVID-19 have demonstrated a strong antibody response in breast milk following maternal infection with SARS-CoV-2 [14,53]. However, to date, no review has comprehensively and in an integrated way discussed a range of studies examining the influence of SARS-CoV-2 infection or vaccination on the composition of various constituents in breast milk, such as cytokines, chemokines, cells, and specific anti-SARS-CoV-2 immunoglobulins. Given this scenario, a comprehensive analysis of these findings is essential to reinforce the importance of maternal vaccination and breastfeeding promotion policies in COVID-19 and encourage future investigations into the immunological protection afforded to infants. Therefore, this review aims to provide an overview of the impact of maternal exposure to SARS-CoV-2 through natural infection and/or vaccination on the immunological composition of breast milk based on the existing studies on the topic.

## 2. Presence of Viral RNA and Anti-SARS-CoV-2 Immunoglobulins in Breast Milk

Early in the pandemic, COVID-19 raised numerous concerns about its transmissibility, including the possibility of vertical transmission via breastfeeding [54]. Although early studies reported the presence of SARS-CoV-2 ribonucleic acid (RNA) in the milk of women with COVID-19 [55,56,57,58], more recent studies have not observed the presence of this virus [59,60,61]. A systematic review of 82 studies on the topic showed that of the 66 studies examining the presence of SARS-CoV-2 in breast milk samples from breastfeeding mothers with confirmed COVID-19, viral RNA was found in 17 of the 38 studies. However, only two studies monitored the activity of the virus, revealing that no infectious virus could be cultured in breast milk samples [62].

This discrepancy in results is likely due to several methodological limitations, including the lack of access, especially early in the pandemic, to targeted and validated protocols for detecting SARS-CoV-2 and aspects related to the collection and handling of breast milk. Breast milk is subdivided into colostrum, transitional, and mature milk based on the time elapsed since delivery (Figure 4). Colostrum has a high concentration of immunological components, decreasing over time and stabilizing mature milk [63]. In studies that analyzed breast milk immunological components in the context of COVID-19, collection time points varied according to study design and maternal exposure to SARS-CoV-2. Samples were generally collected at different stages of maternal infection (acute phase, convalescence, and after recovery) and at specific intervals following vaccination (before vaccination, days or weeks after the first and second dose, and, in some cases, after the booster dose).

Sample storage followed standardized protocols to preserve immunological components. McGuire et al. published a study outlining the best practices for collecting, handling, and storing breast milk in COVID-19 research [64]. Generally, milk was stored at −20 °C or −80 °C for later analysis, while some samples were processed immediately to prevent the degradation of sensitive cells and proteins. The main analytical methods included cytokine and chemokine quantification by ELISAs (Enzyme-Linked Immunosorbent Assays) or Multiplex assays, detection of SARS-CoV-2-specific antibodies by ELISA, and characterization of immune cells in milk by flow cytometry. Additionally, studies that analyzed the presence of SARS-CoV-2 in breast milk samples used reverse transcription–polymerase chain reaction (RT-PCR), which is a rapid and sensitive method that is considered the gold standard for the clinical diagnosis of COVID-19 [65,66] and is also widely used to analyze viral contamination of food [67]. Bäuerl et al. developed and validated a specific protocol to isolate and detect SARS-CoV-2 RNA in human whole milk samples, targeting the N1 region of the nucleocapsid gene and the envelope (E) gene. In their study, all 72 breast milk samples analyzed were negative for the presence of SARS-CoV-2 RNA [60].

What is known to date is that no replication-competent viruses have been found in breast milk samples, suggesting that the viral particles found in breast milk are not infectious [57,62]. Thus, there is no evidence that SARS-CoV-2 is transmitted through breast milk. In summary, scientific evidence indicates a strong antibody response induced after maternal infection with SARS-CoV-2. Fox et al. published one of the first studies on this topic, demonstrating significant levels of SARS-CoV-2-specific IgA in all breast milk samples obtained from donors who had recovered from COVID-19. Their study also showed that 80% of breast milk samples were specifically reactive against the RBD of the SARS-CoV-2 S protein, a critical neutralization epitope. Furthermore, 67% of the samples exhibited immunoglobulin G (IgG) and/or immunoglobulin M (IgM) binding to the virus’s RBD [53].

Pace et al. also demonstrated a rapid, robust, and at least 2-month-long SARS-CoV-2-specific anti-RBD IgA response in breast milk. Their longitudinal study included breastfeeding mothers diagnosed or tested for COVID-19 (including asymptomatic cases) who provided milk samples in the first week of diagnosis and in weeks 2, 3, 4, and 8. The longitudinal analysis revealed that 92% of the breastfeeding mothers produced anti-RBD SARS-CoV-2 IgA in their milk up to day 19, with concentrations increasing during the first weeks after the onset of symptoms or, if asymptomatic, after the day of diagnostic testing [68].

Another study by Fox et al. confirmed that anti-SARS-CoV-2 S protein IgA antibodies present in breast milk are predominantly secretory and can persist for up to 10 months after maternal infection [69]. SIgA antibodies can recognize a wide range of microorganisms, including respiratory viruses such as SARS-CoV-2, influenza, and respiratory syncytial virus, and are key components of mucosal immunity [70]. SIgA is the first line of defense against different antigens, preventing microorganisms from adhering to and penetrating the epithelium, a mechanism known as immunological exclusion. The other functions of SIgA include maintaining gastrointestinal homeostasis, which supports the development of immunological tolerance to commensal organisms, as well as modulating inflammatory reactions, particularly on the mucosal surfaces of the lungs [71].

Interestingly, anti-SARS-CoV-2 IgA is also found in the saliva of infants but only in those who are breastfed, suggesting that breast milk can stimulate the infant mucosal immune response. In a study by Conti et al., IgA antibodies against the SARS-CoV-2 S protein were more concentrated in infants’ saliva than in the breast milk of women who tested positive for COVID-19 at delivery. Their study also showed that breast milk contains IgA anti-S protein immune complexes (also known as antigen–antibody complexes), with concentrations higher in colostrum than in mature milk collected 2 months after delivery [72]. Immune complexes play regulatory roles in the immune system and can actively stimulate the immune response of newborns, preparing them for defense against pathogens recently encountered by the mother. They have also been proposed for the development of therapeutic and preventive vaccines, as well as for stimulating B and T cell responses [73].

Although IgA/SIgA are the predominant antibodies in breast milk, anti-SARS-CoV-2 IgM/SIgM and IgG are also present in breast milk samples. Peng et al. identified the presence of anti-SARS-CoV-2 N protein IgM in 21 of 38 samples collected from eight breastfeeding mothers with a confirmed COVID-19 diagnosis between 3 and 68 days after symptom onset. They demonstrated that most of the IgM-positive samples were collected within the first 40 days from the onset of symptoms of SARS-CoV-2 infection, compared with the IgM-negative samples. In their study, breast milk samples were negative for anti-SARS-CoV-2 IgG even several weeks after infection [59]. Regarding the functions of IgM and IgG present in breast milk and IgA/SIgA, they also act in the defense of the infant intestinal mucosa [74].

In a prospective multicenter longitudinal study, the response of anti-SARS-CoV-2 immunoglobulins was analyzed as a function of time from the diagnosis of COVID-19. The study found a similar proportion of samples positive for IgA anti-SRBD (72.9%) and IgM anti-SRBD (72.9%) in milk samples from breastfeeding mothers infected with or recovered from COVID-19. The positivity rate for IgA anti-SRBD remained relatively stable over time (65.2–87.5%), while most samples positive for IgM anti-SRBD were detected when those collected occurred 11 to 20 days after the diagnosis of COVID-19 (83.3%), with a reduction in their levels to 62.5%. For the positivity rate of IgG anti-SRBD, there was a continuous increase from 47.8% to 87.5% from day 41 to day 206 after confirmation of the diagnosis of COVID-19. There were no time-dependent quantitative differences in endpoint titers for the different breast milk anti-SARS-CoV-2 antibody subtypes, likely due to the high inter- and intra-individual variability identified in the study [60].

In general, the positivity threshold for anti-SARS-CoV-2 antibodies in breast milk samples is determined by using negative controls (pre-pandemic or negative samples) by calculating the mean plus one or more standard deviations (commonly 2 or 3) to establish the minimum positive value. Standard curves and cut-off values established by commercial or internally validated assays may also be used to quantify antibody concentrations. For ELISAs, an optical density (OD) value above a defined cut-off typically indicates positivity [60]. Limits of detection (LOD) and limits of quantification (LOQ) establish the minimum analyte concentration that can be reliably measured. Additionally, ROC (receiver operating characteristic) curve analysis is commonly used to define and validate cut-off points in serological tests, such as ELISAs, and is considered the optimal method for determining thresholds for anti-SARS-CoV-2 IgG, IgM, and IgA antibodies [75].

Understanding the temporal profile of the different classes of antibodies produced after SARS-CoV-2 infection is essential for the interpretation and clinical application of the results found [76,77] (Figure 5). SARS-CoV-2 infection or vaccination commonly induces a classic viral response in serum. Immunoglobulin M is produced first (around 7 to 8 days after infection), followed by IgA, which peaks 2–3 weeks after symptom onset before declining. IgG antibodies appear later in the immune response (they do not develop until around 14 days after infection) but remain detectable for several months after infection and/or vaccination [76,78]. However, in the context of COVID-19, serum anti-RBD IgA has been detected before the appearance of IgM in individuals infected with SARS-CoV-2 [78]. It is worth noting that neonates born to women infected with and/or vaccinated against SARS-CoV-2 during pregnancy present IgG titers of maternal origin (transferred through the placenta) against the SARS-CoV-2 S protein, which can persist up to 6 months of life [79,80].

The recently published cohort study by Fernández-Buhigas et al. also managed to find the three main subtypes of SARS-CoV-2 anti-S1RBD antibodies—IgA (79.6%), IgG (3.1%) and IgM (19.7%)—in colostrum samples from women with laboratory-confirmed SARS-CoV-2 infection during pregnancy (any trimester), delivery, or immediately postpartum (not vaccinated against COVID-19). In their study, no colostrum sample was found to contain all three immunoglobulin subtypes simultaneously. To demonstrate the evolution of antibodies over time, the authors also analyzed transitional and mature milk samples collected from day 7 to 6 weeks postpartum. In the samples, IgG anti-S1RBD and IgA anti-S1RBD were found in 3.23% and 27.42%, respectively. Additionally, the study demonstrated that anti-SARS-CoV-2 immunoglobulins present in colostrum do not appear to vary significantly concerning the gestational trimester in which the infection occurred during pregnancy or their presence in maternal blood [61].

Different studies have reported varying findings, suggesting that the time since the onset of COVID-19 symptoms may or may not affect the transfer of antibodies to breast milk [81,82,83]. A prospective cohort study analyzed the concentration and duration of breast milk antibodies 3 and 6 months after delivery concerning the trimester of maternal SARS-CoV-2 infection. It demonstrated the impact of time since infection and antibody titers in breast milk. They found a higher concentration of anti-S1RBD IgA of SARS-CoV-2 in the breast milk of infected women (not vaccinated against COVID-19) in the second trimester of pregnancy (36%, 3 months after delivery; 38%, 6 months after delivery) and not in those infected in the third trimester (20%, 3 months after delivery; 28%, 6 months after delivery) [81].

Wachman et al. also identified higher concentrations of anti-SARS-CoV-2 S protein IgA in colostrum samples from breastfeeding mothers who had COVID-19 in the first and, more predominantly, in the second trimester of pregnancy compared with those who had the infection in the third trimester [83]. Conversely, Hochmayr et al. identified higher concentrations of anti-SARS-CoV-2 S1RBD IgA and IgG in mature milk samples from women with active COVID-19 in the peripartum period when compared with those with COVID-19 during pregnancy [82].

Unlike IgA/SIgA and IgM/sIgM, the most abundant antibody isotypes in breast milk, low titers of IgG are commonly found in breast milk. In this perspective, some studies failed to detect anti-SARS-CoV-2 IgG in the milk of breastfeeding mothers infected with SARS-CoV-2 [14,84], while others detected low titers [72,85]. Conti et al. detected IgG specific for the S protein of SARS-CoV-2 in low concentration in colostrum and mature milk samples. In the cohort, anti-SARS-CoV-2 IgA antibodies were detectable in all breast milk samples, with a higher concentration in colostrum samples compared to mature milk collected 2 months later [72].

A cohort study that analyzed 141 colostrum samples identified IgM and IgG anti-SARS-CoV-2 N protein in only 7.5% and 3.0% of the milk samples, respectively. According to the authors, the COVID-19 profile of the study population (only mild symptoms) and the diagnosis of the infection (predominantly in the third trimester of pregnancy) could explain the low rate of positive antibodies detected in colostrum [86]. In contrast, Fox et al. identified IgG anti-SARS-CoV-2 S protein in 75% of breastfeeding mothers who recovered from COVID-19 (asymptomatic cases or those who presented mild to moderate symptoms), with 13% presenting high titers [54]. Pace et al. also demonstrated the presence of IgG specific for different SARS-CoV-2 antigens in 80% of breast milk samples after COVID-19 diagnosis [54].

The factors influencing breast milk antibody titers are not yet fully understood. From this perspective, some studies have analyzed whether SARS-CoV-2-specific antibody profiles in breast milk may manifest differently based on the severity of COVID-19. Bäuerl et al. and Pace et al. found no statistically significant differences between the SARS-CoV-2 anti-RBD IgA response in symptomatic and asymptomatic cases of COVID-19 [54,60].

On the other hand, Pullen et al., when analyzing the serological profile of colostrum samples from breastfeeding mothers previously infected with SARS-CoV-2 during pregnancy, showed that women who had more severe COVID-19 transferred higher levels of IgG and IgA binding to the Fc receptor against several SARS-CoV-2 specificities in breast milk. In comparison, women with less severe disease transferred higher levels of functional antibodies, namely, NK cell-activating antibodies and nucleocapsid-specific antibodies that induce phagocytosis of monocytes and neutrophils. Their study classified breastfeeding mothers into four groups based on the National Institutes of Health COVID-19 severity criteria [85].

These findings may suggest that women with more severe disease and potentially more inflammatory serological profiles transfer higher titers of neutralizing antibodies and less functional antibodies to breast milk [85]. In the context of infectious diseases, an antibody’s function is related to its biological effect on a pathogen or its toxin. In this sense, functional antibodies are antibodies that, in addition to binding to their antigen (neutralizing function), perform other active biological functions, which include neutralization of infectivity, phagocytosis, antibody-dependent cellular cytotoxicity, and complement-mediated lysis of pathogens or infected cells [87]. Sheehan et al. also demonstrated that in addition to neutralizing activity, breast milk antibodies mediate phagocytic activity directed toward the SARS-CoV-2 virus, which corroborates the maternal transfer of functional antibodies to the newborn [88].

A prospective study that analyzed antibodies in breast milk samples at different stages of lactation from breastfeeding mothers with a history of SARS-CoV-2 infection during pregnancy or active peripartum infection (vaccinated or not against COVID-19) also demonstrated variations in breast milk antibody responses concerning disease severity. Breastfeeding mothers with moderate to severe COVID-19 have higher concentrations of IgA and IgG anti-S1RBD of SARS-CoV-2 in transitional milk when compared to asymptomatic breastfeeding mothers or those with mild symptoms of the infection. Significantly higher IgA and IgM anti-S1RBD concentrations were also observed in mature milk from breastfeeding mothers with moderate to severe COVID-19, suggesting that the severity of SARS-CoV-2 infection may influence antibody titers in breast milk [82].

In the abovementioned study, the researchers classified maternal SARS-CoV-2 infection according to the criteria defined by the Society for Maternal–Fetal Medicine as mild disease when it included flu-like symptoms (fever, cough, myalgia, and anosmia) and moderate disease in the presence of lower respiratory tract disease with dyspnea, pneumonia, refractory fever, and abnormal blood gases (oxygen saturation < 93%). Severe disease was defined as a respiratory rate greater than 30 breaths per minute and the need for oxygen supplementation, while critical disease was defined as multiple organ failure, respiratory failure requiring high-flow nasal cannula therapy, or mechanical ventilation [89].

Although the severity of COVID-19 is likely to influence the levels and types of antibodies in breast milk, further studies are needed to draw definitive conclusions [59]. However, there is evidence of a positive association between high antibody titers and increased clinical severity of COVID-19 for systemic antibodies [76].

The longitudinal study published by Yang et al., in addition to confirming the presence of antibodies specific for SARS-CoV-2 in breast milk for up to 12 months after COVID-19, demonstrated that, unlike the declining humoral blood response, there is little change in anti-SARS-CoV-2 antibody titers in breast milk over time. In their study, at the end of 12 months, half of the breastfeeding mothers who recovered from COVID-19 (not vaccinated against the disease) exhibited a less than 2-fold reduction in anti-SARS-CoV-2 S protein IgA levels [90]. Pullen et al. also demonstrated a highly stable transfer of anti-SARS-CoV-2 IgM, IgA, and IgG into breast milk despite the expected loss of IgM in maternal serum. This persistent IgM response in breast milk, even after the loss of serum IgM response to natural infection, may reflect the continued production of secretory IgM in breast milk during lactation [85].

The persistent presence of anti-SARS-CoV-2 IgA, IgM, and IgG in breast milk after episodes of natural maternal infection suggests that anti-SARS-CoV-2 antibodies in breast milk may reflect either long-lived plasma cells in the gut-associated lymphoid tissue (GALT) and/or mammary gland or result from continuous antigen stimulation in these compartments by other human coronaviruses, or even from repeated exposures to SARS-CoV-2 [85,90]. Cross-reactive immune responses have already been observed in individuals without previous exposure to SARS-CoV-2. This is due to the high homology of SARS-CoV-2 with other coronaviruses (α-coronavirus and β-coronavirus) [91].

In this sense, Demers-Mathieu et al. evaluated whether the potential protective effects of breast milk antibodies could also decrease the risk of viral infection by other coronaviruses. Their study analyzed whether breast milk antibodies against the S1 and S2 subunits of SARS-CoV-2 cross-react with the S1 and S2 subunits of other coronaviruses (HCoV-OC43 and HCoV-229E) in breastfeeding mothers with a positive PCR test for COVID-19, in breastfeeding mothers with a history of viral symptoms during the COVID-19 pandemic, and unexposed breastfeeding mothers (pre-pandemic). They showed that IgG levels against the S2 subunit of SARS-CoV-2 were higher in the PCR-positive and COVID-19 viral symptom groups than in the unexposed group (pre-pandemic), with no differences in SIgA/IgA and SIgM/IgM. This lack of difference between PCR-positive, virally symptomatic, and unexposed groups for SIgA/IgA and SIgM/IgM reactive to SARS-CoV-2 may be related to their polyreactive ability to bind to different epitopes [92].

These findings reflect the cross-reactive capacity of breast milk antibodies that can recognize a wide range of human coronaviruses, including the so-called common cold coronaviruses [91]. Since the S proteins of SARS-CoV-2 and other human coronaviruses share structural similarities, immunoglobulins produced by B cells upon antigen recognition could cross-react with these coronaviruses. Thus, breast milk antibodies against SARS-CoV-2 may provide additional immune defense to infants and reduce the risk of COVID-19 infection [92]. From this perspective, it can be suggested that cross-reactive T and B cells are also present in breast milk, as cross-reactive immunity is mediated not only by antibodies but also by memory cells, which are triggered by a specific pathogen or antigen and can respond to other pathogens or antigens as well [93].

Furthermore, it is worth discussing that the vaccination of pregnant and lactating women against SARS-CoV-2 also induces the production and release of specific antibodies in breast milk. However, studies indicate that the humoral response of breast milk in women with natural infection differs from that of women who received protection via a vaccine [59,94,95,96]. Furthermore, data from a systematic review indicate that, unlike natural infection, vaccination does not appear to induce a significant increase in secretory antibody titers, and the IgA present in milk after immunization appears to be almost exclusively of systemic rather than mucosal origin [59].

Also, according to their systematic review, Dimitroglou et al. identified that vaccination against COVID-19 induces the secretion of anti-protein S antibodies in breast milk, while natural infection mainly induces antibodies against the N protein of SARS-CoV-2. Higher IgG titers are observed in the milk of vaccinated women compared to those previously infected with the virus. The data for IgA are still conflicting. And for IgM, vaccination does not seem to influence its levels [59].

Young et al. compared the temporal breast milk IgA and IgG responses and microneutralization activity against SARS-CoV-2 between breastfeeding mothers with PCR-confirmed COVID-19 and breastfeeding mothers vaccinated with Pfizer-BioNTech (BNT162b2) or Moderna (mRNA-1273) up to 90 days after infection or vaccination. This cohort demonstrated that natural infection was associated with a robust and rapid breast milk IgA response that remained stable up to 90 days after diagnosis. In contrast, vaccination was associated with a more uniform IgG-dominant response, with concentrations increasing after each vaccine dose and beginning to decline 90 days after the second dose. Neutralizing activity was identified in breast milk from both the infected and vaccinated groups, with slightly higher activity in the naturally infected group [95].

Regarding differences in the immune response according to vaccine type [97,98] (Figure 6), a cohort study that analyzed 1887 breast milk samples collected over 70 days from women who received one of four different SARS-CoV-2 vaccines—BNT162b2 or mRNA-1273, or AZD1222 or Ad26.COV2.S—showed that the breast milk antibody response differed between vaccines during the study period. Overall, maternal vaccination during lactation with an mRNA-based vaccine resulted in higher IgA and IgG antibody responses against SARS-CoV-2 in breast milk compared to vector-based vaccines. In summary, after immunization with vector-based vaccines, IgA levels were not detectable in breast milk at the group level. IgG levels only became detectable after vaccination with the AZD1222 vaccine [99].

mRNA and adenoviral vector vaccines have distinct compositions and mechanisms of action, which may result in different immune profiles in breast milk. mRNA vaccines use lipid nanoparticles to deliver the mRNA encoding the S protein, inducing a strong humoral response and stimulating the production of antibodies, mainly IgA and IgG. This approach leads to the rapid activation of innate and adaptive pathways, resulting in high levels of these antibodies that can effectively protect the infant. In contrast, adenoviral vector vaccines use a modified virus to deliver the S protein gene, activating the cellular immune response more strongly and stimulating antibody production [97,98]. These differences in antibody composition and quantity may directly affect the efficacy of passive immunity transmitted through breast milk, offering varying levels of protection to the infant.

Selma-Royo et al. also demonstrated that anti-SARS-CoV-2 IgA and IgG levels in breast milk were dependent on the vaccine type, with higher IgG and IgA levels in mRNA-based vaccines when compared to AstraZeneca, and on previous exposure to the virus. In their study, all breast milk samples from breastfeeding mothers who received the COVID-19 vaccine remained positive for anti-SARS-CoV-2 IgG at the end of the vaccination cycle (two vaccine doses). In contrast, only 10–70% of the samples were classified as positive for IgA [94].

A recently published systematic review also showed that mRNA vaccines have a higher rate of IgA and IgG positivity in breast milk after vaccination when compared to vector-based vaccines, suggesting that mRNA vaccines may offer infants a higher level of protection via anti-SARS-CoV-2 antibodies [99]. Vector-based vaccines may have reduced efficacy in cases where the host has already been exposed to the virus used as a vector and has produced neutralizing antibodies against it before the vaccine is administered. In this sense, the viral vector used in AstraZeneca’s AZD1222 vaccine, for example, comes from a more common type of adenovirus known to infect a larger population, which may require more doses of the vaccine or a greater need for booster doses [97].

Another systematic review with meta-analysis also confirmed that vaccinating pregnant and lactating women may positively affect the mother–child dyad since it increases immunoglobulin levels in breast milk, providing passive immunization to the infant. In the review, the levels of anti-SARS-CoV-2 IgA and IgG in breast milk increased with subsequent doses of the COVID-19 vaccine. A positive relationship was found for the second (coefficient = 0.91 for IgA and coefficient = 1.77 for IgG) and third (coefficient = 1.23 for IgA and coefficient = 3.73 for IgG) vaccine doses. There was a significant increase in anti-SARS-CoV-2 IgG in breast milk, mainly after the second dose of the vaccine. The authors found that the Moderna vaccine had a strong positive relationship with IgA (coefficient = 6.22) and IgG (coefficient = 6.65) anti-SARS-CoV-2 antibodies in breast milk, followed by the Pfizer vaccine (coefficient = 1.41 and 3.03). The Johnson & Johnson vaccine had a weak positive relationship for IgA (coefficient = 0.39) and a negative relationship for IgG (coefficient = −0.33) [100].

An important point to be discussed is that to perform their functions, breast milk immunoglobulins need to survive the actions of digestive proteases along the infant’s gastrointestinal tract to their site of action [101]. Typically, the gastric juice of infants is acidic (pH > 3.0 under feeding conditions) and contains only pepsin, a lipase enzyme. In contrast, their duodenal juice is more alkaline, with bile salts and enzymes to digest proteins, fats, and carbohydrates. In adults, the gastric pH is much lower and much more acidic, in addition to the difference in gastric and intestinal enzyme concentration and activity levels [102]. Based on these characteristics, it is expected that much of the immunoglobulin ingested by an infant is partially or fully digested; however, it is assumed that some portion of the antibody remains intact or partially intact, maintaining the ability to bind to an antigen. Furthermore, immunoglobulins are generally relatively more resistant to gastrointestinal digestion than other proteins in colostrum and breast milk [103]. In summary, studies indicate that the stability of breast milk immunoglobulins during gastric digestion is greater in preterm infants than in full-term infants, probably due to the greater gastric pepsin activity and proteolysis in full-term infants [101].

To demonstrate the potential protective role of anti-SARS-CoV-2 antibodies in breast milk after maternal vaccination against COVID-19, studies have investigated the survival of specific antibodies through a static in vitro digestion protocol [104,105]. Studying digestion using in vivo assays, particularly in humans, is complex and involves technical, ethical, and financial constraints [106]. Thus, in vitro alternatives, such as digestive models, simulate the physiological conditions of the human gastrointestinal tract, allowing the study of the digestibility and structural changes in specific foods or components ingested less expensively, with easy sampling accessibility and without involving ethical issues, when compared to in vivo models [102,106].

An in vitro static digestion protocol for full-term newborns (28 days of life), developed from parameters based on in vivo data published in the literature, was proposed by Ménard et al. [106] and used in the pilot study by Calvo-Lerma et al. [105]. In their pilot study, they demonstrated, through simulated in vitro digestion, that anti-SARS-CoV-2 IgA and IgG antibodies in breast milk from women infected and vaccinated against COVID-19 can survive digestion, with levels remaining above cut-off values even after the intestinal phase of digestion [105].

Pieri et al. analyzed the presence of IgA, IgG, and SIgA in milk samples from breastfeeding mothers who received at least two doses of an mRNA-based vaccine (Pfizer/BioNTech, Moderna) or an adenovirus-based vaccine (AstraZeneca). Their study also demonstrated, through a static in vitro digestion protocol, that although slightly reduced, IgA antibodies produced after vaccination resisted the gastric and intestinal phases of infant digestion, while IgG was more prone to degradation in both phases of digestion. Specifically, in the gastric phase of digestion, there was degradation of 17.1%, 8.6%, and 3.2% of the IgG, IgA, and SIgA classes, respectively. While in the intestinal phase of digestion, there was 74.3%, 14.3%, and 48.4% degradation of the IgG, IgA, and SIgA antibodies, respectively. Antibody concentrations in breast milk were analyzed before and after a 1-h simulated gastric digestion and a 2-h simulated infant intestinal digestion protocol [104].

These studies provide valuable insights into the potential protective role of the SARS-CoV-2 vaccine for infant health, suggesting that vaccine-induced antibodies may be immunologically active during infant digestion and thus exert their protective effects in the infant intestinal lumen during breastfeeding [104,105]. Furthermore, the study by Pieri et al. also demonstrated that anti-SARS-CoV-2 SIgA is present in the milk of women vaccinated against COVID-19 and that it resists the gastric phase of digestion, although it shows some reduction during the intestinal phase [104].

Finally, based on the results presented, it is evident that both natural infection and vaccination against SARS-CoV-2 induce a robust maternal immune response based on specific IgA/SIgA, IgM, and IgG immunoglobulins with neutralizing and functional capacity, which may or may not vary depending on the severity or presence of symptoms and the time of diagnosis of COVID-19. Although it is not possible to say precisely when anti-SARS-CoV-2 antibodies can be detected in breast milk after maternal infection and/or vaccination against COVID-19, which class of antibodies appears first, and for how long they persist in breast milk, evidence suggests that the IgA class is the most prevalent type in milk samples after natural infection. At the same time, IgG is found at higher levels in the milk of vaccinated women, particularly after the second and third doses of the vaccine. Additionally, the levels of anti-SARS-CoV-2 IgA and IgG in breast milk depend on the type of vaccine, with mRNA vaccines inducing higher titers than vector-based vaccines. Well-conducted randomized clinical trials are essential to examine the duration of antibody persistence in breast milk and the long-term effect of the vaccine.

Additional studies are needed to demonstrate the protective effect of antibodies against SARS-CoV-2 infection in infants of vaccinated and/or infected mothers. However, evidence suggests that anti-SARS-CoV-2 antibodies can be transferred to infants via breast milk, protecting them against COVID-19 and other coronaviruses. It has been demonstrated that anti-SARS-CoV-2 antibodies in breast milk can bind to the viral surface and block its replication cycle, potentially protecting the infant from infections [67]. Therefore, given the potential for transmission of passive immunity to the infant through breast milk, breastfeeding should be encouraged in both contexts of SARS-CoV-2 infection or vaccination against COVID-19, under the recommendations of the Centers for Disease Control and Prevention and the World Health Organization, that support the maintenance of breastfeeding during and after COVID-19 [65,107].

## 3. Cytokines, Chemokines, Growth Factors, and Cellular Components in Breast Milk in the Context of COVID-19

It is known that COVID-19 induces an increase in inflammatory cytokines in the blood of individuals infected with SARS-CoV-2 [108,109,110]. A prospective case–control study showed that women with symptomatic COVID-19 at any point during the third trimester of pregnancy had higher levels of the growth factors EGF, G-CSF, and HGF, the pro-inflammatory cytokines IL-1β and IL-6, and the chemokines IP-10/CXCL10, MCP-1, and MIG in peripheral blood samples collected shortly after delivery compared to asymptomatic cases [111]. However, to date, few studies have investigated the impact of COVID-19 on the profile of cytokines and other immunological compounds, such as chemokines, growth factors, and milk cells, in women infected with or recovered from SARS-CoV-2 infection [112].

Narayanaswamy et al. were the first to describe the cytokine profile in the colostrum of women with COVID-19. In their study, breastfeeding mothers symptomatic for SARS-CoV-2 infection showed higher mean levels for all cytokines analyzed (IFN-γ, TNF-α, IL-1β, IL-2, IL-4, IL-6, IL-10, IL-12, and IL-13), except for IL-8. The levels of IFN-γ, IL-4, IL-6, and IL-12 were significantly higher in the colostrum of the breastfeeding mothers compared to the asymptomatic mothers [113].

Colostrum generally presents a more pro-inflammatory cytokine profile than mature milk [114]. In the context of COVID-19, studies have shown that SARS-CoV-2 infection can induce specific and distinct inflammatory responses in the body [108,109,110]. Pro-inflammatory cytokines, such as IL-1α, IL-1β, IL-12, IFN-γ, and TNF-α, are involved in the upregulation of inflammatory reactions. In contrast, anti-inflammatory cytokines, such as IL-4, IL-10, IL-13, and IL-1Ra, help counteract these effects by downregulating the pro-inflammatory response. Together, these cytokine classes are crucial in regulating the host’s response to infection and inflammation [115], as they are key mediators facilitating communication between immune cells [116].

Higher levels of pro-inflammatory cytokines can benefit newborns at the beginning of lactation, when the neonatal intestine is more immature, contributing to the defense of the intestinal mucosa and the development of the infant’s immune system [117]. Due to protease inhibitors in breast milk, such as α1-antichymotrypsin and α1-antitrypsin, which protect proteins against digestion, milk cytokines and chemokines can reach the neonatal intestine intact. Furthermore, during the first 3 months of life, gastric digestion of proteins is reduced due to the limited secretion of pepsin and hydrogen ions and the general immaturity of the newborn’s digestive capacities, which also favors the bioactive effects of the different constituents of breast milk on neonatal health [118].

A multicenter, prospective, case–control study determined the concentration of cytokines, chemokines, and growth factors in breast milk samples from 37 cases (women with full-term pregnancies and confirmed non-severe COVID-19) and 45 controls (SARS-CoV-2-negative women under identical conditions) using consecutive samples collected at 1 and 5 weeks postpartum. Cytokine concentrations (IFN-γ, IL-1Ra, IL-4, IL-6, IL-9, IL-13, and TNF-α) were higher in the breast milk from the breastfeeding mothers with symptomatic and asymptomatic COVID-19 compared with the control group at both sampling times. Cytokine concentrations were significantly higher in the breast milk samples from the breastfeeding mothers whose RT-PCR remained positive at 3 weeks postpartum compared with the control group. Chemokines (eotaxin, IP-10, MIP-1α, and RANTES/CCL5) and growth factors (fibroblast growth factor—FGF, granulocyte–macrophage colony-stimulating factor—GM-CSF, IL7, and platelet-derived growth factor-BB—PDGF-BB) also showed higher concentrations in the breast milk from the breastfeeding mothers with COVID-19 than in the control group at 1 week postpartum. Overall, the concentrations of immunological compounds decreased over time, particularly in the milk samples from the control group, and the severity of the disease (symptomatic or asymptomatic COVID-19) did not affect the immunological profile of breast milk [112].

According to García et al., the concentrations of most of the immunological factors analyzed in their study remained stable over time in milk samples from breastfeeding mothers with COVID-19. In contrast, most of these compounds decreased significantly from the first to the fifth week postpartum in breastfeeding mothers without the disease. Although this may be a physiological response of the mother to the infection, such results may also suggest that during COVID-19, the immunological profile of breast milk adapts to provide protection to the infant against maternal infection. These results also suggest that the immune system of infected breastfeeding mothers reacted efficiently against SARS-CoV-2, producing a distinct pattern of cytokines, chemokines, and growth factors in breast milk, which persisted over time [112].

Trofin et al. also quantified 10 cytokines and chemokines in breast milk from 22 breastfeeding mothers infected with SARS-CoV-2 and 26 breastfeeding mothers vaccinated against COVID-19. The analyses were compared with milk samples from 10 breastfeeding mothers without a history of COVID-19 and who were unvaccinated (negative anti-SARS-CoV-2 IgG). The milk samples were collected 30 and 60 days after the booster dose of Pfizer or Moderna vaccine and after the onset of COVID-19 symptoms. The concentrations of TNF-α, IL-6, IFN-β, IL-10, IL-1β, IFN-γ, IL-2, GM-CSF, and IL-5 were within the reference range in the breast milk from breastfeeding mothers infected or vaccinated with SARS-CoV-2. All breastfeeding mothers excreted IP-10 in high concentrations in breast milk, with no differences between groups. In the study, no significant differences were recorded between the concentrations of cytokines and chemokines in the breast milk samples among the three groups evaluated. However, the levels of TNF-α and IL-2 were higher in the breastfeeding mothers with COVID-19 than in the vaccinated breastfeeding mothers or in the control group but still within the reference range. The vaccine, SARS-CoV-2 infection, the age of the breastfed infant, parity, and maternal age were factors that interfered with the variation in cytokine concentrations in the study [119].

Cells present in breast milk, as well as mammary gland epithelial cells during lactation, actively participate in the production of IL-1β, IL-6, and TNF-α, which can respond to additional stimuli after leaving the breast. The functional significance of the different cytokines in breast milk in vivo is still unknown. However, pro-inflammatory cytokines present in breast milk, such as IL-1β, IL-6, and TNF-α, may provide innate immune responses in defense against enteric or respiratory pathogens, triggering a systemic inflammatory response, while anti-inflammatory factors, such as IL-10 and TGF-β may help modulate cytokine responses to infection, facilitating immune defense and minimizing tissue damage [114].

A study that analyzed how the side effects of mRNA vaccines against COVID-19 correlate with the cytokine response induced by vaccination in breast milk demonstrated that the milk of women immunized with the first and second doses of the COVID-19 vaccine had significantly higher levels of IFN-γ compared to breast milk provided before vaccination. Furthermore, breastfeeding mothers who reported experiencing side effects after vaccination against SARS-CoV-2 had higher levels of IFN-γ in the milk provided after the second dose compared to the milk provided before receiving the vaccine. According to the authors, this elevation of IFN-γ in the milk of women receiving the mRNA vaccine may protect infants against several viral infections of the respiratory tract, including SARS-CoV-2 [120].

Corroborating this finding, it has been demonstrated that the production of IFN-γ coincides with the onset of protective immunity to SARS-CoV-2 during infection or after vaccination against COVID-19 [121,122]. IFN-γ (type II IFN) is produced by NK cells and macrophages (effector cells in innate immunity), as well as T helper 1 (Th1)-type CD4^+^ T lymphocytes and CD8^+^ T cells that participate in the adaptive response [123], being one of the most important and vital pro-inflammatory cytokines for the body’s defense against viral infections [110]. In the context of COVID-19, an in vitro study demonstrated that IFN-γ can inhibit the replication cycle of SARS-CoV-2 in human epithelial cells through the generation of nitric oxide, which is an essential biological mediator in the immune system with broad antimicrobial activity against intracellular pathogens [124].

To test the hypothesis that milk from breastfeeding mothers with COVID-19 has higher levels of cytokines related to Th cells (IL-4, IL-17, IFN-γ, IL-2, and TNF-β) than milk from breastfeeding mothers without SARS-CoV-2 infection, Demers-Mathieu et al. analyzed 40 milk samples from 10 breastfeeding mothers with a COVID-19 diagnosis confirmed by RT-PCR, 10 breastfeeding mothers with viral symptoms suggestive of COVID-19, and 20 breastfeeding mothers without infection. In their study, the level of IL-2 in breast milk was higher in breastfeeding mothers without infection than in those with confirmed COVID-19 but did not differ from the group with viral symptoms. IL-4, TNF-β, IFN-γ, and IL-7 were comparable between the three groups. IL-2, IL-4, IL-17, and TNF-β levels decreased with increasing time from viral symptoms to milk collection [125].

Regarding these results, it is known that IL-2 is the primary inducer of Th1 cells, essential for mediating antiviral defense and adaptive immunity [125]. Th cells (CD4^+^ T lymphocytes) are central in orchestrating adaptive immune responses and act in defense against infections, coordinating the actions of other cells. Their developmental process allows CD4^+^ cells to recognize a wide range of foreign antigens without provoking self-reactivity to self-antigens. Based on the expression of cytokines and transcription factors, Th cells are categorized into different cell subsets, which include Th1 (IFN-γ and T-bet), Th2 (IL-4, IL-5, IL-13, and GATA3), Th17 (IL-17, IL-22, and RORγt), and regulatory T (Tregs) (IL-10, TGF-β, IL-35, and Foxp3) cells [126].

In this sense, it has been suggested that the level of IL-2 in breast milk was affected by the presence and type of maternal infection. It is speculated that the transfer of IL-2 from breast milk to infants may stimulate the growth and development of T cells. However, further studies are needed to identify the role of IL-2 during neonatal development and its role in preventing viral infections. Regarding the lack of difference in the levels of IL-4, IL-17, IFN-γ, and TNF-β between infected and uninfected groups, the authors suggest that it may be associated with the time elapsed between infection and milk collection since cytokines may disappear after a few months of activation of the immune response [125].

Concerning the analysis of breast milk cells in the context of SARS-CoV-2 infection, the first published work was a case report that analyzed samples of mature breast milk expressed from a breastfeeding mother before (1 month after delivery/before the pandemic) and during COVID-19 (7 months after delivery). The study found no significant difference between the cellular components (T and B cells, NK, innate lymphoid cells (ILCs), neutrophils, macrophages and dendritic cells) in breast milk before and during infection; however, there was a qualitative change in the percentage of macrophages expressing IFNα (a type I interferon) in the breast milk sample with active COVID-19, which showed an 8-fold increase compared to the milk collected before infection [127].

According to the authors, it is likely that immune cells in the breast produce high levels of IFNα due to active systemic COVID-19 infection. In this sense, infants of mothers infected with SARS-CoV-2 may benefit from the increase in IFNα due to its role as a natural antiviral, which may contribute to some level of protection against COVID-19 [55]. Type I IFNs (α/β/ω) are key components of the immediate antiviral response, capable of restricting viral replication and spread. The deficiency of an adequate type I IFN response, such as IFNα, is observed in severe patients with COVID-19. Thus, IFNα/β are potential therapeutics for treating SARS-CoV-2 infection [128,129].

Except for the previously described case report, Narayanaswamy et al. were the first to comprehensively analyze the cellular response to SARS-CoV-2 infection in breast milk. Cryopreserved cells from fresh mature milk samples collected approximately 35 days and 4 months after COVID-19 diagnosis were analyzed by flow cytometry. Their study demonstrated that the distribution of CD4^+^ and CD8^+^ T cells was similar in all milk samples, regardless of the collection period. The authors identified that T cells in the milk of women infected with SARS-CoV-2 presented high levels of the mucosal localization marker, with higher expression of CD103 on CD8^+^ T cells compared to CD4^+^ T cells. This high expression of CD103 on breast milk T cells suggests that they originate from a tissue-resident population [130]. Although the origin of breast milk T cells remains unclear, human and animal studies indicate that T cells in breast milk originate from both the gut and maternal blood [40,131].

Additionally, breast milk from women recovered from COVID-19 was observed to be enriched with antigen-experienced T cells, evidencing that most of the breast milk CD4^+^ and CD8^+^ populations were central memory T cells (CD45RO^+^/CCR7^+^), regardless of the time of milk expression. Interestingly, the percentage of effector memory T cells (CD45RO^+^/CCR7^-^) in the CD4^+^ and CD8^+^ populations increased significantly over the 4-month follow-up [130].

These results suggest that the milk of breastfeeding mothers recovered from SARS-CoV-2 infection presents a T cell profile with cytotoxic and memory phenotypes. The fact that effector memory and mucosal T cells are transferred to the infant via breast milk strengthens the role of breastfeeding in transferring passive protection to infants against SARS-CoV-2 [130].

Effector CD4^+^ T cells produce cytokines that activate other immune cells, such as macrophages and CD8^+^ T lymphocytes, killing infected cells and helping B cells produce antibodies to mediate humoral immune responses. CD8^+^ T lymphocytes, also called cytotoxic T lymphocytes, act to eliminate intracellular pathogens and can kill cells carrying the target antigen by releasing cytotoxic molecules (such as granzymes and perforin) and secreting cytokines, such as IFN-γ and TNF-α [132].

Memory T lymphocytes may protect the newborn through previous maternal immunological experience. They also provide high levels of cytokine-producing helper and cytotoxic T cells, which may help in the postnatal development of infant immune responses. Thus, breastfeeding in early life acts as a compensatory mechanism between the mother–child dyad since breast milk can help infants build their immune defense [133].

Our research group also performed a comprehensive analysis of the immunological, biochemical, and cellular contents in the colostrum of breastfeeding mothers with a history of symptomatic and asymptomatic COVID-19 during pregnancy. In that study, breastfeeding mothers who manifested symptoms of SARS-CoV-2 infection during pregnancy produced colostrum with a higher proportion of the two NK cell subtypes, CD3^−^CD56^dim^CD16^+^CD27^−^ and CD3^−^CD56^bright^CD16^−^CD27^+^IFN^−^γ^+^, associated with a lower concentration of the cytokines IFN-α2 and GM-CSF, when compared to the colostrum of breastfeeding mothers asymptomatic for the infection. Although the real impact of such changes in colostrum on neonatal health is not known, the study demonstrated that mild symptomatic COVID-19 infection during pregnancy caused significant changes in the cell and cytokine profile of the colostrum of women not vaccinated against COVID-19, which may have a direct impact on the infant’s innate immunity and intestinal development [134].

NK cells are present in breast milk at different stages of lactation [135]. They participate in the innate immune response, acting as the first line of defense against infection. They can directly kill target cells and interact with both antigen-presenting cells and T cells [36]. It is speculated that in breast milk, NK cells may play a critical role in maintaining mucosal homeostasis and inducing infant immunological tolerance. In humans, NK cells can be divided into two subsets—CD56^dim^ NK cells (cytotoxic) and CD56^bright^ NK cells (cytokine producers) [136]. In the context of COVID-19, evidence suggests that SARS-CoV-2 infection may affect the tissue distribution and effector functions of NK cells and that a rapid NK cell response may determine a good clinical outcome in patients with COVID-19 [137].

It is known that mRNA vaccines induce spike protein-reactive B and T cells in the blood of vaccinated individuals, including pregnant and lactating women [138,139]. A study that sought to evaluate the effects of mRNA vaccines on the immunological composition of breast milk demonstrated that spike protein-specific CD4^+^ T cells are transferred to breast milk after vaccination and can pass long-lasting immunity to the breastfed infant. Additionally, the study demonstrated that breastfeeding mothers have a higher frequency of circulating RBD^+^ memory B cells and higher titers of anti-RBD antibodies. It was also shown that SIgA is produced early (after 10 days of vaccination) by the mammary mucosa in response to the first dose of the mRNA vaccine [140].

Gonçalves et al. also attempted to demonstrate the presence of RBD-reactive B cells in breast milk, but this was not possible due to the limited number of B cells detected in the samples, possibly due to the low volume of milk collected. Since memory T cells are long-lived, the authors suggested that such findings open the possibility that protection transferred by breast milk may still be present in the infant, even after weaning. Since CD4^+^ T cells are crucial in mediating mRNA vaccine protection, it is possible that spike-reactive T cells transferred to breast milk may mediate protection against infections through dissemination in the upper respiratory tract and infant gut [140].

Like T cells, B lymphocytes act as an important component of the adaptive immune system. As the most important effector cells in humoral immunity, B cells can transform into plasma cells and then produce antibodies [141]. In breast milk, B lymphocytes differ from their blood counterparts. While in blood there is a predominance of naïve B cells, in breast milk, B cells are predominantly class-switched memory cells (CD27^+^IgD^−^), which express IgG or IgA surface molecules, with phenotypic characteristics of activated cells, expressing high levels of CD38 and low expression of complement receptors, which are characteristic of plasma cells [142].

Breast milk-derived B lymphocytes also present a specific profile of mucosal adhesion molecules (CD44^+^, CD62L^−^, α4β7^−/+^, α4β1^+^), suggesting that these cells originate in the mammary gland but resemble the B cells associated with the GALT, which also express the α4 and β7 integrin chains and do not express CD62L. These findings support the concept that the mammary gland functions as an effector site of the mucosal immune system [142]. However, the migration of B cells to the mammary glands is also mediated by the chemokine CCL28, which favors the accumulation of IgA-secreting cells during lactation [143].

Continuing the analysis of breast milk cells in the context of SARS-CoV-2 infection, a cohort study compared the phenotype, diversity, and antigen specificity of T cells in breast milk and peripheral blood of breastfeeding mothers vaccinated with SARS-CoV-2 mRNA. The study included breastfeeding mothers who received two doses of the BNT162b2 or mRNA1273 vaccines, had no history of COVID-19, and were seronegative for the SARS-CoV-2 N protein. It was demonstrated that compared to blood, breast milk contained higher frequencies of effector (CD45RO^+^/CCR7^-^) and central memory (CD45RO^+^/CCR7^+^) T cells that expressed mucosal localization markers (CCR9^+^ and CD103^+^). In breast milk, T cell receptors specific to the SARS-CoV-2 S protein were also more frequent compared to blood [144].

The cohort further showed that these specific anti-spike cells in breast milk respond in vivo to antigen restimulation as their population expanded in breast milk after the third dose of the mRNA vaccine. These results indicate that the lactating breast represents a key site of mucosal immunity, in addition to containing a distinct population of SARS-CoV-2-specific T cells that can be modulated by maternal vaccination, with potential implications for passive protection of the infant against COVID-19 [144].

Regarding the transfer of immune cells from breast milk to the infant, it is expected that in the first weeks of life, the gastric and intestinal environment of the newborn is exceptionally hospitable for maternal cells and other components of breast milk to survive the conditions of the upper digestive tract and be able to cross the neonatal intestinal barrier. From this perspective, one of the most widely accepted theories is that of paracellular passage, where breast milk cells cross the weakened interepithelial junctions (tight junctions) of the still-immature intestine of the infant [42].

According to the results presented (Figure 7), the analysis of the impact of COVID-19 on the immunological composition of breast milk revealed distinct alterations in the cytokines and T cells of infected or recovered breastfeeding mothers, which may impact neonatal protection against SARS-CoV-2. Breastfeeding mothers with symptomatic or asymptomatic COVID-19 presented higher levels of several cytokines, such as IFN-γ, IL-4, IL-6, and TNF-α, compared to control groups. Both natural infection and vaccination against COVID-19 appear to influence the concentration of immunological compounds in breast milk, with higher levels of different classes of IFN (IFN-γ, IFN-α). CD4^+^ and CD8^+^ T cells from the milk of women exposed to SARS-CoV-2 present memory characteristics and mucosal localization, suggesting a role of passive protection for the infant. mRNA vaccination against SARS-CoV-2 also induces a T cell response specific to the virus’s S protein, which may provide long-lasting immunity to the infant.

Furthermore, SARS-CoV-2 infection appears to alter the macrophage profile, while symptomatic COVID-19 during pregnancy appears to influence colostrum composition by changing the proportion of NK cells. Moreover, the severity of COVID-19 does not seem to substantially affect the immunological concentrations of milk, suggesting that the immunological profile of breast milk may be influenced mainly by infection or vaccination but not exclusively by the severity of symptoms. Overall, the findings suggest that breast milk from infected or recovered women and women vaccinated against SARS-CoV-2 may be an essential source of protection for infants, particularly those under 6 months of age who cannot be vaccinated against COVID-19.

These findings reinforce the recommendations of the Centers for Disease Control and Prevention (CDC) and align with those of professional medical organizations, including the American College of Obstetricians and Gynecologists, the Society for Maternal–Fetal Medicine, and the American Society for Reproductive Medicine, which advise that all individuals aged 6 months or older receive the updated COVID-19 vaccine, including pregnant and lactating women. Scientific evidence indicates that immunization is safe and effective in these groups, protecting both mother and baby. Vaccination during pregnancy significantly reduces the risk of serious complications associated with SARS-CoV-2 infection while allowing for the transfer of antibodies to the fetus, thus providing additional protection to the newborn. During breastfeeding, antibodies and other bioactive compounds in breast milk offer an extra layer of defense against infection, further emphasizing the importance of immunization during this period. Pregnant and lactating women should discuss vaccination options with their healthcare providers to make informed decisions, considering the proven benefits of vaccine protection and the potential risks of infection [145].

## 4. Conclusions

This comprehensive review demonstrated that natural infection and vaccination against COVID-19 produce a specific humoral and cellular response against SARS-CoV-2, which is passed to the infant through breast milk. This reinforces the importance of breastfeeding from the first hour of the newborn’s life. It was also shown that maternal exposure to SARS-CoV-2 promotes changes in the composition of breast milk, producing a distinct pattern of cytokines, chemokines, and growth factors.

These findings demonstrate the dynamism of breast milk’s composition and its susceptibility to changes due to the mother’s health status and, more specifically, her history of viral infections. Although the real impact of such changes in breast milk on neonatal and infant health is unknown, the results presented demonstrate that symptomatic or asymptomatic infection by COVID-19 or even vaccination against the disease causes significant changes in the profile of bioactive compounds in breast milk, which can directly impact the infant’s immunity and intestinal development. Such changes in breast milk are likely to modulate the infant’s immune response, enhancing defense against COVID-19 and other coronaviruses while promoting immunological tolerance. Additionally, they may influence immune system development and disease susceptibility.

It is worth noting that this dynamism in the composition of breast milk, associated with the complexity and diversity of its constituents, can make it difficult to interpret the findings found in different studies. Furthermore, many methods used to analyze breast milk samples may not be adequately sensitive or specific enough to detect certain immunological constituents or distinguish between different isoforms or variants of proteins and antibodies. Thus, the interpretation of the results found on the impact of maternal exposure to SARS-CoV-2 on the composition of breast milk is still unclear, and the lack of references may make it difficult to compare the results and to assess their clinical relevance to the health of the infant.

However, considering that breast milk reflects maternal and infant health conditions and provides infants with protection against diseases related to intestinal and respiratory infections, it is expected that such changes found in the composition of breast milk are a reflection of the physiological responses of the maternal organism to SARS-CoV-2 infection and/or vaccination against COVID-19, which may contribute to the establishment of immune defense early in the life of newborns. In this perspective, further studies are needed to establish the influence of viral infections and maternal vaccination on the composition of breast milk and other investigations in colostrum, transitional milk, and mature milk from lactating women vaccinated and/or recovered from COVID-19. The impact of changes in breast milk immunological components following maternal exposure to SARS-CoV-2 on immune programming and long-term health consequences in infants should be explored in future studies.

## Figures and Tables

**Figure 1 ijms-26-02600-f001:**
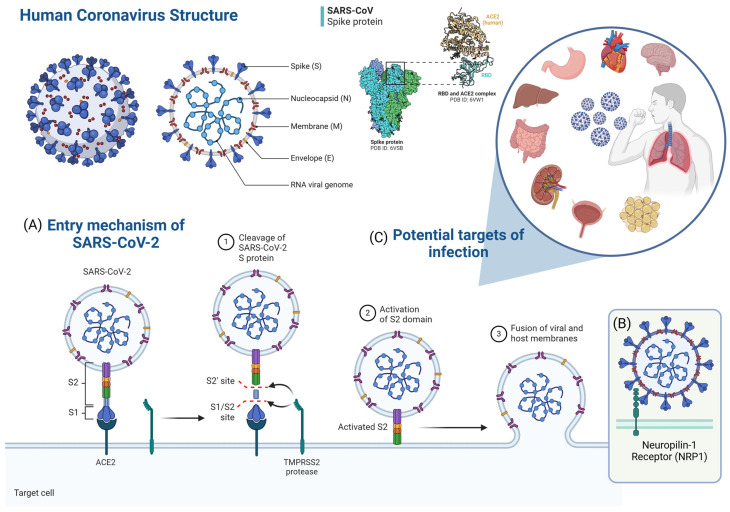
SARS-CoV-2 structure and cellular entry mechanism. (**A**) The S protein of SARS-CoV-2 is composed of two functional subunits, S1 and S2. The S1 subunit contains the receptor-binding domain (RBD), which binds to angiotensin-converting enzyme 2 (ACE2) on the target cell, while the S2 subunit mediates the fusion of viral and host cell membranes. The binding of the S protein to ACE2 on the target cell promotes cleavage of the S1-S2 site by the transmembrane serine protease TMPRSS2, activating the S2 subunit trimers. This fusion process merges the viral and host lipid bilayers, releasing the viral ribonucleoprotein complex into the cell. (**B**) Another entry route for SARS-CoV-2 is the endosome, where cathepsins can cleave the S protein. Other coreceptors, such as neuroligin-1 (NRP1) and proteases (cathepsin L, TMPRSS11D, and TMPRSS13), may also play a role in the entry of SARS-CoV-2 into the host cells. (**C**) ACE2, which is the functional receptor for SARS-CoV-2, is a plasma membrane protein present on the cell surface of several types of human cells. In this way, SARS-CoV-2 can infect the human body through the respiratory tract and spread rapidly to other organs, causing not only severe respiratory complications, but also a variety of damage to other tissues. Adapted from Lamers and Haagmans [4]. Created by the author (2025) using BioRender https://www.biorender.com (accessed on 31 January 2025).

**Figure 2 ijms-26-02600-f002:**
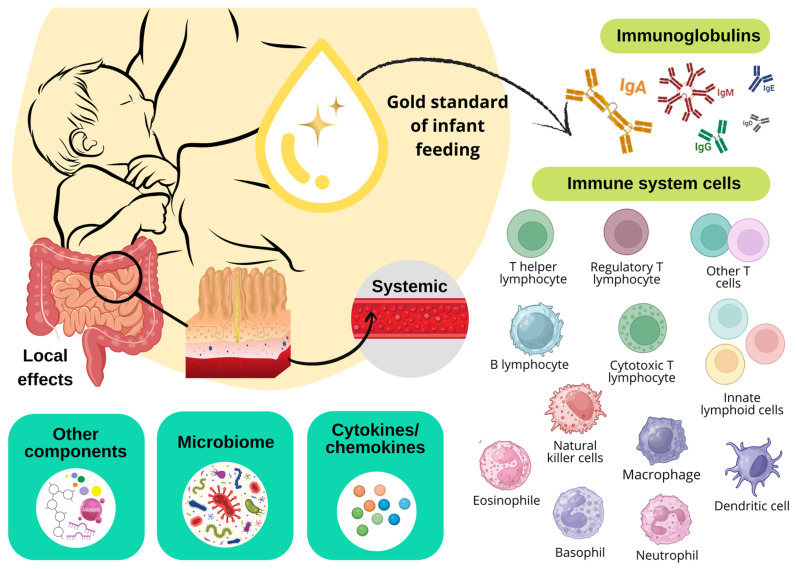
Components with immunological properties present in breast milk. Created by the author (2025) using Canva.com.

**Figure 3 ijms-26-02600-f003:**
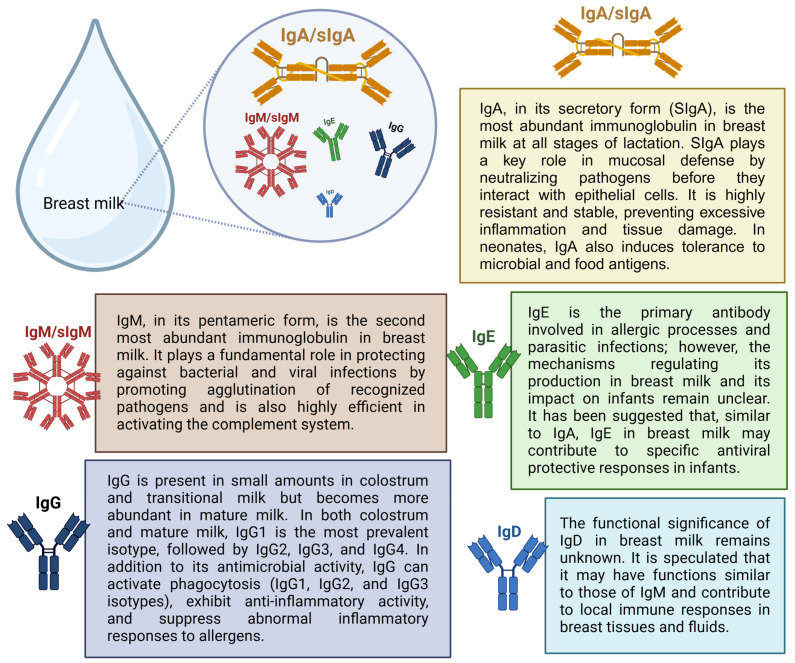
Main functions of breast milk immunoglobulins. Adapted from Rio-Aige [32]. Created by the author (2025) using BioRender.

**Figure 4 ijms-26-02600-f004:**
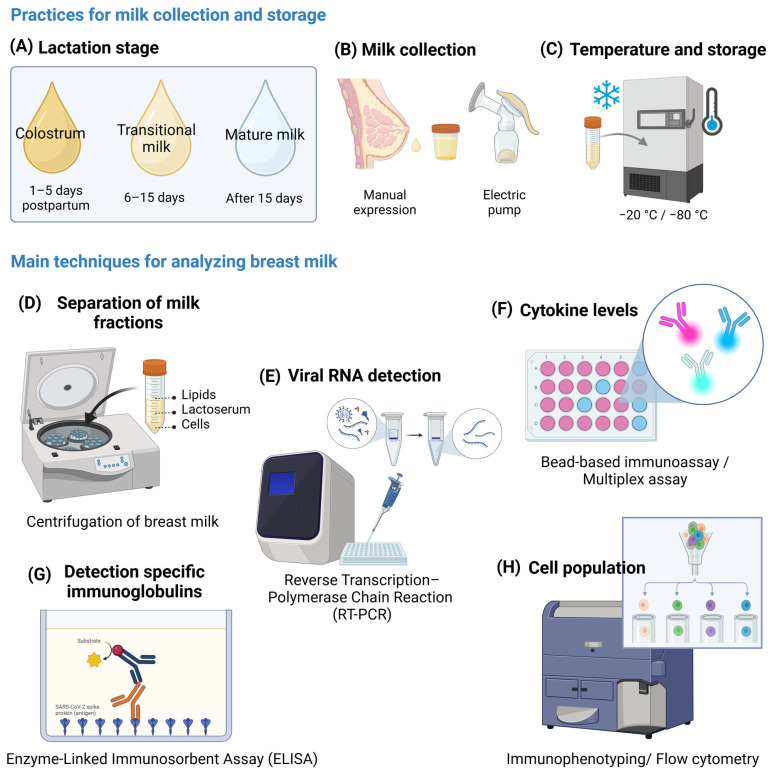
Leading practices for milk collection, storage, and techniques for analyzing breast milk. (**A**) Lactation is a dynamic process that progresses through three phases, impacting breast milk’s appearance, volume, and composition. Colostrum is a thick fluid that is rich in immunological components and proteins. Transitional milk gradually increases energy density, while mature milk has higher lactose and fat concentrations. (**B**) Milk can be collected through hand expression or an electric/manual pump. Pump components can become contaminated and require complete sterilization or disinfection. During manual expressions, hands should be washed, or clean gloves should be worn. The collection container must be sterile. (**C**) Refrigeration, freezing, thawing, and heating can impact the stability of several milk components. It is recommended to store milk at 4 °C for up to 48 h or at −20 °C/−80 °C for at least six months without compromising its immunological properties. (**D**) Milk comprises fractions, such as cells, lipids, and an aqueous phase, which can be separated by centrifugation. (**E**) RT-PCR is the most common technique for identifying SARS-CoV-2 in human fluids, including breast milk. (**F**) Bead-based detection systems allow simultaneous analysis of multiple analytes with minimal sample volumes using coded microspheres and flow cytometry. (**G**) An ELISA is an immunoassay used to detect pathogens by quantifying antibodies that interact with antigens adsorbed on a solid support. (**H**) Immunophenotyping, performed by flow cytometry, categorizes cells based on surface proteins using fluorescent antibodies. Created by the author (2025) using BioRender.

**Figure 5 ijms-26-02600-f005:**
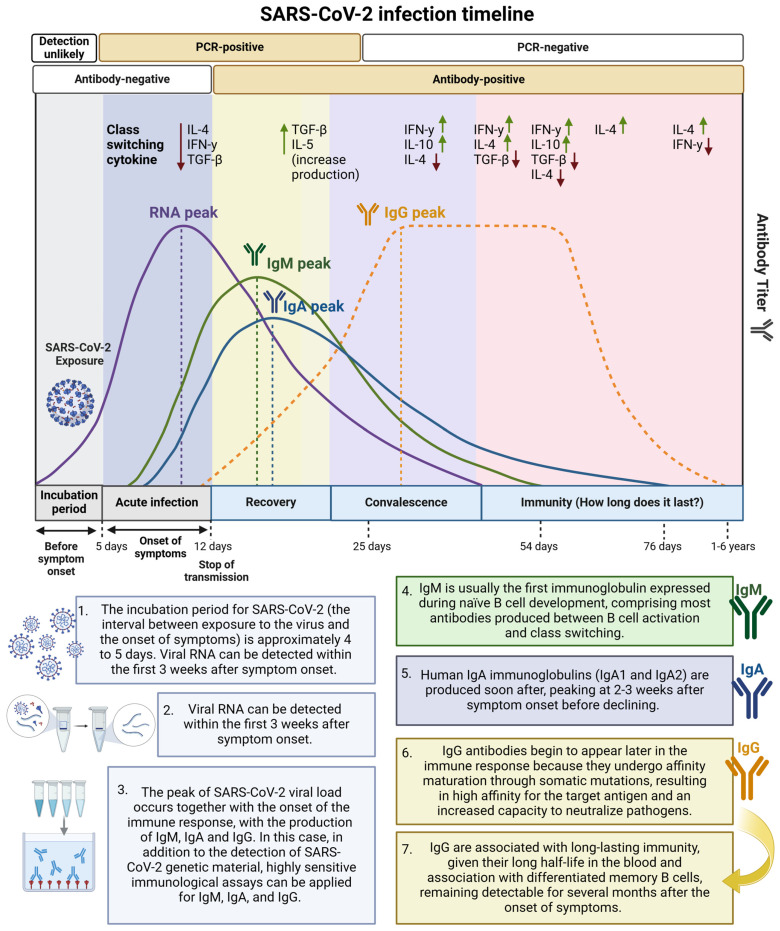
SARS-CoV-2 infection timeline. This figure illustrates the approximate timeline from SARS-CoV-2 infection to immunity. Adapted from Galipeau et al. [76] and Mattioli et al. [77]. Created by the author (2025) using BioRender.

**Figure 6 ijms-26-02600-f006:**
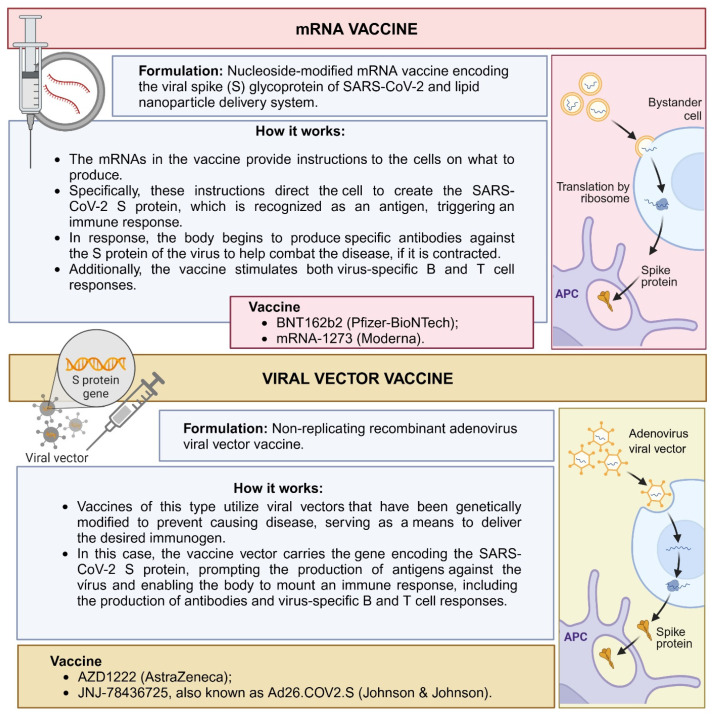
Main types of SARS-CoV vaccines available. Antigen-presenting cell (APC). Adapted from Hendaus and Jomha [81] and Muhar et al. [82]. Created by the author (2025) using BioRender.

**Figure 7 ijms-26-02600-f007:**
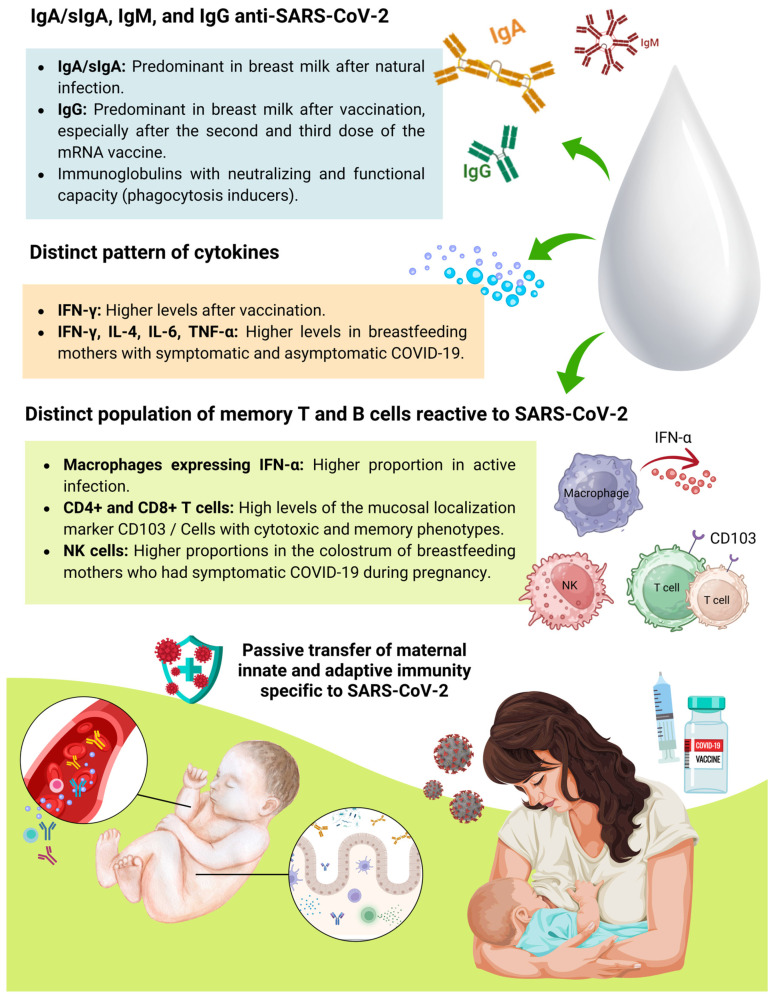
Modifications in the immunological composition of breast milk in response to SARS-CoV-2 infection and vaccination. Breast milk from women infected with COVID-19, recovered from COVID-19, or vaccinated exhibited significant changes in immunological components, potentially protecting infants, particularly those under six months of age who cannot be vaccinated. Locally, antibodies, cytokines, and immune cells can regulate the inflammatory response in the mucosa of the gastrointestinal and respiratory tracts, protecting them against infections and promoting homeostasis. Systemically, antibodies present in milk can be absorbed by the infant, offering protection against pathogens. In addition, cells transferred through breastfeeding reinforce the infant’s immune response, facilitating immunological memory and an efficient adaptive response to future infections. Created by the author (2025) using Canva.com.

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
