# Peer review of "Impact of Maternal Exposure to SARS-CoV-2 on Immunological Components of Breast Milk"

_ijms, 2025, doi:10.3390/ijms26062600_

Round 1

Reviewer 1 Report

Comments and Suggestions for Authors

This article is a review study on the impact of SARS-CoV-2 on breast milk composition, exploring the changes in the immune composition of breast milk after maternal infection or vaccination and its potential protective effect on the infant. The study noted that although no infectious virus was found in breast milk, antibodies against SARS-CoV-2 are present in breast milk and may provide passive immune protection to infants through breastfeeding. In addition, immunologically active substances in breast milk, such as cellular components, cytokines, and chemokines, also change after maternal infection or vaccination, which may significantly affect the development of the infant's immune system and disease defense capabilities. Research shows that breast milk still has important immunoprotective value in the context of the COVID-19 pandemic, and continued breastfeeding is recommended despite maternal infection or vaccination. Although this study demonstrates innovation in analytical methods, some parts still require further improvement. Therefore, it is recommended that the article be accepted with appropriate revisions. Here are some key points for further discussion and consideration:

  1. It is recommended to add information on the role of immune components in breast milk in other infectious diseases. (Page 3, lines 85-86)
  2. The article mentioned that the composition of breast milk will be dynamically adjusted according to the needs of the baby, but did not explore its mechanism in depth. A brief discussion of possible regulatory mechanisms, such as hormone levels and infant feedback signals, is recommended. (Page 3, line 90)
  3. The article mentions analyzing breast milk samples, but does not detail the collection time points, methods, or storage conditions. It is recommended to add a brief description of these aspects. (Page 6, lines 164-165)
  4. This article discusses changes in antibody titers in breast milk, but does not specify the detection criteria for antibody titers (e.g., positivity thresholds, quantitative methods). It is recommended that these standards be included. (Page 8, line 252)
  5. The article states that mRNA vaccines and adenoviral vector vaccines induce different antibody responses in breast milk. Further analysis of potential mechanisms for this difference, such as vaccine composition and immune pathways, is recommended, as well as a discussion of their potential impact on infant protection. (Page 15, lines 449-457)
  6. It is recommended to discuss the impact of the digestive stability of antibodies in breast milk on infant protection. (Page 16, lines 485-486)
  7. Some figures (e.g., Figure 7) have limited explanatory text, which may make it difficult for readers to understand the information conveyed. It is recommended to add detailed explanatory text below the figures to highlight key findings and their significance. (Page 24, line 825)
  8. This article discusses the relationship between antibody titers in breast milk and vaccination but does not provide specific recommendations for vaccination strategies. It is recommended that vaccination recommendations for pregnant and lactating women be provided in the conclusion based on the study results. (Page 25, line 837)
  9. It is recommended to elucidate the long-term effects of changes in the immune composition of breast milk on infant health. (Page 25, lines 838-841)
  10. Some of the references in the article are out of date. It is recommended to add the latest research results, especially on breast milk composition and SARS-CoV-2. (Page 27, line 885)
Comments on the Quality of English Language The English could be improved to express the research more clearly.

Author Response

Open Review

Quality of English Language

(x) The English could be improved to more clearly express the research.
( ) The English is fine and does not require any improvement.

Is the work a significant contribution to the field?

Is the work well organized and comprehensively described?

Is the work scientifically sound and not misleading?

Are there appropriate and adequate references to related and previous work?

Is the English used correct and readable?

Comments and Suggestions for Authors

This article is a review study on the impact of SARS-CoV-2 on breast milk composition, exploring the changes in the immune composition of breast milk after maternal infection or vaccination and its potential protective effect on the infant. The study noted that although no infectious virus was found in breast milk, antibodies against SARS-CoV-2 are present in breast milk and may provide passive immune protection to infants through breastfeeding. In addition, immunologically active substances in breast milk, such as cellular components, cytokines, and chemokines, also change after maternal infection or vaccination, which may significantly affect the development of the infant's immune system and disease defense capabilities. Research shows that breast milk still has important immunoprotective value in the context of the COVID-19 pandemic, and continued breastfeeding is recommended despite maternal infection or vaccination. Although this study demonstrates innovation in analytical methods, some parts still require further improvement. Therefore, it is recommended that the article be accepted with appropriate revisions. Here are some key points for further discussion and consideration:

Answer: We appreciate the opportunity to revise our manuscript and the valuable suggestions provided. Below, we'd like to point out our responses to the comments and the changes made. Blue text indicates the content added to the manuscript. We'll be here for any further clarifications.

  1. It is recommended to add information on the role of immune components in breast milk in other infectious diseases. (Page 3, lines 85-86)

Answer: We appreciate the suggestion. It is indeed interesting to include findings on the role of immunological components of breast milk in other infectious diseases. Additional explanations on the mechanisms by which breast milk protects against infections have been added. We believe this information enriches the introduction and reinforces the importance of immunological mechanisms in infant protection.

‘Breast milk protects against infectious through multiple mechanisms, including strengthening the infant's epithelium barrier via growth factors, transferring antimicrobial agents like lactoferrin, lysozyme, and oligosaccharides; promoting beneficial microbial growth, and conveying maternal antigen-specific immunity through lymphocytes and antibodies (Verhasselt et al., 2024). It is documented that the composition of breast milk dynamically adjusts to infections, enhancing the infant's immune defenses (Vassilopoulou et al., 2024). Studies have shown that maternal and/or infant infections rapidly increase leukocyte levels in breast milk, which return to baseline after recovery (Riskin et al., 2012; Hassiotou et al., 2013). Zheng et al. reported that infant respiratory infections promote the migration of anti-inflammatory macrophages into breast milk, suggesting an additional mechanism of infant protection (Zheng et al., 2020). Chemokines and their receptors play an essential role in innate immunity against viral infections (Bosire et al., 2007; Farquhar et al., 2005). Farquhar et al. found that higher concentrations of macrophage inflammatory protein-1β (MIP-1β) and stromal cell-derived factor-1α (SDF-1α) in breast milk were linked to a lower risk of vertical human immunodeficiency virus 1 (HIV-1) transmission, while elevated RANTES (regulated upon activation, expressed, and secreted by normal T cells) levels were associated with increased risk, regardless of viral RNA levels in breast milk (Farquhar et al., 2005).”

  1. The article mentioned that the composition of breast milk will be dynamically adjusted according to the needs of the baby, but did not explore its mechanism in depth. A brief discussion of possible regulatory mechanisms, such as hormone levels and infant feedback signals, is recommended. (Page 3, line 90)

Answer: We appreciate the comment and acknowledge the relevance of the topic. However, we believe that an in-depth discussion of the regulatory mechanisms of breast milk composition goes beyond the scope of this study. To address the recommendation without deviating from the central focus of the review, we have included a brief mention of the possible factors and mechanisms influencing milk composition, without delving too deeply into the subject. We believe this approach sufficiently presents the topic while maintaining the manuscript’s objectivity.

“The composition of breast milk varies according to the stage of lactation (colostrum, transitional milk, and mature milk) and between full-term and preterm newborns. Factors such as ethnicity, diet, maternal age, parity, maternal and child health status, environment, and milk management (collection, storage, and pasteurization) also influence its composition. Its regulation is achieved through a complex interplay of hormonal mechanisms and feedback signals from the infant, who actively contributes to adjusting milk composition through both mechanical and biochemical cues (Ballard; Morrow, 2012).”

  1. The article mentions analyzing breast milk samples, but does not detail the collection time points, methods, or storage conditions. It is recommended to add a brief description of these aspects. (Page 6, lines 164-165)

Answer: We appreciate your feedback. In response, we have included a brief description of the collection sites, methods, and storage conditions for breast milk samples. Figure 4 (“Main practices for milk collection, storage, and techniques for analyzing breast milk”) has been modified to incorporate this information. We hope this addition addresses the recommendation and enhances the clarity of the methods used. Although this information is valuable, we believe that a more in-depth discussion would exceed the scope of this manuscript.

“Breast milk is subdivided into colostrum, transitional milk, and mature milk based on the time elapsed since delivery (Figure 4). Colostrum has a high concentration of immunological components, which decrease over time and stabilize in mature milk (Lokossou et al., 2022). In studies that analyzed breast milk immunological components in the context of COVID-19, collection time points varied according to study design and maternal exposure to SARS-CoV-2. Samples were generally collected at different stages of maternal infection (acute phase, convalescence, and after recovery) and at specific intervals following vaccination (before vaccination, days or weeks after the first and second dose, and, in some cases, after the booster dose). Sample storage followed standardized protocols to preserve immunological components. McGuire et al. published a study outlining best practices for collecting, handling, and storing breast milk in COVID-19 research. Generally, milk was stored at −20 °C or −80 °C for later analysis, while some samples were processed immediately after to prevent the degradation of sensitive cells and proteins. The main analytical methods included cytokine and chemokine quantification by ELISA (Enzyme-Linked Immunosorbent Assays) or Multiplex assays, detection of SARS-CoV-2-specific antibodies by ELISA, and characterization of immune cells in milk by flow cytometry.”

  1. This article discusses changes in antibody titers in breast milk, but does not specify the detection criteria for antibody titers (e.g., positivity thresholds, quantitative methods). It is recommended that these standards be included. (Page 8, line 252)

Answer: We appreciate the recommendation to include detection criteria for antibody titers. In response, we have added details on the standards used to determine antibody positivity in breast milk. We believe this inclusion clarifies the quantitative methods used and fulfills the recommendation.

“In general, the positivity threshold for anti-SARS-CoV-2 antibodies in breast milk samples is determined using negative controls (pre-pandemic or negative samples) by calculating the mean plus one or more standard deviations (commonly 2 or 3) to establish the minimum positive value. Standard curves and cut-off values established by commercial or internally validated assays may also be used to quantify antibody concentrations. An optical density (OD) value above a defined cut-off for ELISA typically indicates positivity (47). Limits of detection (LOD) and quantification (LOQ) establish the minimum concentration analyte concentration that can be reliably measured. Additionally, ROC (receiver operating characteristic) curve analysis is commonly used to define and validate cut-off points in serological tests, such as ELISA. It is considered the optimal method for determining thresholds for anti-SARS-CoV-2 IgG, IgM, and IgA antibodies (Oluka et al., 2023).”

  1. The article states that mRNA vaccines and adenoviral vector vaccines induce different antibody responses in breast milk. Further analysis of potential mechanisms for this difference, such as vaccine composition and immune pathways, is recommended, as well as a discussion of their potential impact on infant protection. (Page 15, lines 449-457)

Answer: Thank you for your comment. Figure 6 (“Main types of SARS-CoV vaccines available”) provides some differentiation between the two vaccine types. We have added a brief analysis of the potential mechanisms underlying the differences in antibody responses induced by mRNA and viral vector vaccines. We believe this inclusion addresses the recommendation and enhances the discussion on the impact of these vaccines on infant protection.

“mRNA vaccines and adenoviral vector vaccines have distinct compositions and mechanisms of action, which may result in different immune profiles in breast milk. mRNA vaccines use lipid nanoparticles to deliver the mRNA encoding the S protein, inducing a strong humoral response and stimulating the production of antibodies, mainly IgA and IgG. This approach leads to rapid activation of both innate and adaptive pathways, resulting in high levels of these antibodies that can effectively protect the infant. In contrast, adenoviral vector vaccines use a modified virus to deliver the S protein gene, activating the cellular immune response more strongly while also stimulating antibody production (82,83).  These differences in antibody composition and quantity may directly affect the efficacy of passive immunity transmitted through breast milk, offering varying levels of protection to the infant.”

  1. It is recommended to discuss the impact of the digestive stability of antibodies in breast milk on infant protection. (Page 16, lines 485-486)

Answer: We appreciate the comment. However, we believe that the impact of digestive stability of antibodies in breast milk on infant protection is already adequately addressed in the manuscript. Therefore, we consider an additional discussion unnecessary to maintain the central focus of the study.

“An important point to be discussed is that to perform their functions, breast milk immunoglobulins need to survive the actions of digestive proteases along the infant's gastrointestinal tract to their site of action (86). Typically, the gastric juice of infants is acidic (pH >3.0 under feeding conditions) and contains only pepsin, a lipase enzyme. In contrast, duodenal juice is more alkaline, with bile salts and enzymes to digest proteins, fats, and carbohydrates. In adults, the gastric pH is much lower and much more acidic, in addition to the difference in gastric and intestinal enzyme concentration and activity levels (87). Based on these characteristics, it is expected that much of the immunoglobulin ingested by the infant is partially or fully digested; however, it is assumed that some portion of the antibody remains intact or partially intact, maintaining the ability to bind to an antigen. Furthermore, immunoglobulins are generally relatively more resistant to gastrointestinal digestion than other proteins in colostrum and breast milk (88). In summary, studies indi-cate that the stability of breast milk immunoglobulins during gastric digestion is greater in preterm infants than in full-term infants, probably due to the greater gastric pepsin activity and proteolysis in full-term infants (86).”

  1. Some figures (e.g., Figure 7) have limited explanatory text, which may make it difficult for readers to understand the information conveyed. It is recommended to add detailed explanatory text below the figures to highlight key findings and their significance. (Page 24, line 825)

Answer: We appreciate your feedback. In response, we have included a caption in Figure 7 that highlights the key findings and their significance. We believe the explanatory text in the caption sufficiently clarifies the information for readers.

  1. This article discusses the relationship between antibody titers in breast milk and vaccination but does not provide specific recommendations for vaccination strategies. It is recommended that vaccination recommendations for pregnant and lactating women be provided in the conclusion based on the study results. (Page 25, line 837)

Answer: Important remark. However, as this is a literature review, we did not consider it appropriate to include specific vaccination recommendations in the conclusion. Instead, we added information on vaccination recommendations for pregnant and lactating women in the section preceding the conclusion, based on the results of the reviewed studies. We believe this approach aligns with the scope of the manuscript and adequately reflects the available evidence.

“These findings reinforce the recommendations of the Centers for Disease Control and Prevention (CDC) and align with those of professional medical organizations, including the American College of Obstetricians and Gynecologists, the Society for Maternal-Fetal Medicine, and the American Society for Reproductive Medicine, which advise that all individuals aged 6 months or older receive the updated COVID-19 vaccine, including pregnant and lactating women. Scientific evidence indicates that immunization is safe and effective in these groups, protecting both mother and baby. Vaccination during pregnancy significantly reduces the risk of serious complications associated with SARS-CoV-2 infection while allowing for the transfer of antibodies to the fetus, thus providing additional protection to the newborn. During breastfeeding, antibodies and other bioactive compounds present in breast milk offer an extra layer of defense against infection, further emphasizing the importance of immunization during this period. Pregnant and lactating women should discuss vaccination options with their healthcare providers to make informed decisions, considering the proven benefits of vaccine protection and the potential risks of infection (CDC, 2024).”

  1. It is recommended to elucidate the long-term effects of changes in the immune composition of breast milk on infant health. (Page 25, lines 838-841)

Answer: We appreciate your comment. Although the long-term effects of changes in the immune composition of breast milk on infant health have not yet been fully clarified, we note that the observed changes may modulate the infant’s immune response, enhance defense against COVID-19 and other coronaviruses, and influence immune system development and disease susceptibility. We recognize that further research is needed to elucidate these long-term effects.

“Such changes in breast milk are likely to modulate the infant's immune response, enhancing defense against COVID-19 and other coronaviruses while promoting immunological tolerance. Additionally, they may influence immune system development and disease susceptibility.”

“The impact of changes in breast milk immunological components following maternal exposure to SARS-CoV-2 on immune programming and long-term health consequences in infants should be explored in future studies.”

  1. Some of the references in the article are out of date. It is recommended to add the latest research results, especially on breast milk composition and SARS-CoV-2. (Page 27, line 885)

Answer: Thank you for your comment. Please note that the review covers work carried out from the beginning of the pandemic to the present, and we have incorporated more robust references identified in our search for recent research. If you identify any additional relevant references that are missing, we would be grateful if you could let us know so we can include them.

_____________________________________________________________________________

Comments on the Quality of English Language

The English could be improved to express the research more clearly.

The English has been improved.

Submission Date

01 February 2025

Date of this review

11 Feb 2025 15:12:01

Reviewer 2 Report

Comments and Suggestions for Authors

The document is well-written, interesting, and covers a very important topic, but it has some minor issues. After carefully reviewing the manuscript, it can be accepted after the authors make the following minor corrections:

  1. COVID-19 is no longer as alarming or dangerous as it once was. The authors should provide a stronger rationale for the importance of their study and justify why this research remains relevant today.
  2. The paper lacks of clear research gap and justification

 3. I might have missed something, but assessement on inflammatory cytokines, chemokines, and cellular responses lacks a logical flow, making it difficult to follow how these immune components interact within the broader context of COVID-19 and breastfeeding

4. Can the authors say more on the variants and their impact on immune response

5. Authors must also present a very strong view, on what is being added or new, compared to the following papers,

Centeno-Tablante E, Medina-Rivera M, Finkelstein JL, Rayco-Solon P, Garcia-Casal MN, Rogers L, Ghezzi-Kopel K, Ridwan P, Peña-Rosas JP, Mehta S. Transmission of SARS-CoV-2 through breast milk and breastfeeding: a living systematic review. Ann N Y Acad Sci. 2021 Jan;1484(1):32-54.

Florea RM, Sultana CM. COVID-19 and breastfeeding: can SARS-CoV-2 be spread through lactation? Discoveries (Craiova). 2021 Jun 30;9(2):e132. doi: 10.15190/d.2021.11. PMID: 34754901; PMCID: PMC8570917.

Detection of SARS-CoV-2 in Milk From COVID-19 Positive Mothers and Follow-Up of Their Infants

Author Response

Open Review

Quality of English Language

( ) The English could be improved to more clearly express the research.
(x) The English is fine and does not require any improvement.

Is the work a significant contribution to the field?

Is the work well organized and comprehensively described?

Is the work scientifically sound and not misleading?

Are there appropriate and adequate references to related and previous work?

Is the English used correct and readable?

Comments and Suggestions for Authors

The document is well-written, interesting, and covers a very important topic, but it has some minor issues. After carefully reviewing the manuscript, it can be accepted after the authors make the following minor corrections:

  1. COVID-19 is no longer as alarming or dangerous as it once was. The authors should provide a stronger rationale for the importance of their study and justify why this research remains relevant today.

Answer: We appreciate your comment. Although the severity of COVID-19 has diminished compared to the early stages of the pandemic, our study remains relevant for several reasons. First, it provides valuable insights into the impacts of maternal COVID-19 infection and vaccination on the immune composition of breast milk. Second, understanding how maternal exposure to SARS-CoV-2 influences passive immunity in infants can inform public health strategies and clinical practices, especially as new variants emerge. Finally, our findings contribute to the broader understanding of viral infections and immune modulation in breast milk, with potential implications for future pandemics and other infectious diseases. Additional information has been added to underscore the importance of our study.

“Although SARS-CoV-2 infection has reached an endemic phase with periodic outbreaks, population immunity acquired through natural infection and/or vaccination declines over time, increasing the risk for vulnerable groups (Razonable et al., 2024). Additionally, vaccine booster uptake has been decreasing. As of March 2024, only 23% of adults in the United States reported receiving the updated 2023–2024 COVID-19 vaccine (CDC, 2024). Between October 2023 and April 2024, approximately 70% of pregnant women had not received this vaccine before or during pregnancy (Katherine et al., 2024). In this context, achieving adequate vaccination coverage among pregnant and lactating women remains a global challenge (Comparcini et al., 2024).”

  1. The paper lacks a clear research gap and justification.

Answer: We appreciate the feedback. In response, we revised sections of the article to address the research gap. Specifically, we emphasize that a comprehensive analysis of these findings is essential to highlight the importance of maternal vaccination and breastfeeding promotion policies in the context of COVID-19 and to encourage future investigations into the immunological protection afforded to infants.

“Given this scenario, a comprehensive analysis of these findings is essential to reinforce the importance of maternal vaccination and breastfeeding promotion policies in the context of COVID-19 and encourage future investigations into the immunological protection afforded to infants.”

  1. I might have missed something, but assessment on inflammatory cytokines, chemokines, and cellular responses lacks a logical flow, making it difficult to follow how these immune components interact within the broader context of COVID-19 and breastfeeding.

Answer: We appreciate the feedback. To ensure clarity, we organized the manuscript by first addressing inflammatory cytokines and chemokines, followed by cellular responses. This sequential structure provides a logical flow of information on how these immune components interact in the broader context of COVID-19 and breastfeeding.

  1. Can the authors say more about the variants and their impact on the immune response?

      Answer: Thank you for your feedback. We have included information on variants and their impact on the immune response in response. We believe this inclusion addresses your request.

“Over time, numerous SARS-CoV-2 mutant variants have been identified, with the Variants of Concern (VOC) – Alpha (lineage B.1.1.7), Beta (lineage B.1.351), Gamma (lineage P.1), Delta (lineage B.1.617.2) and Omicron (lineage B.1.1.529) – standing out due to their increased transmissibility and/or virulence potential (Cascella et al., 2023). The primary concern regarding the emergence of new SARS-CoV-2 variants lies in the reduced effectiveness of vaccines and natural immunity, driven by genomic alterations, particularly in the coding regions of the S protein, which increase viral fitness compared to ancestral strains (Mistry et al., 2022).”

  1. Authors must also present a very strong view, on what is being added or new, compared to the following papers, 

Centeno-Tablante E, Medina-Rivera M, Finkelstein JL, Rayco-Solon P, Garcia-Casal MN, Rogers L, Ghezzi-Kopel K, Ridwan P, Peña-Rosas JP, Mehta S. Transmission of SARS-CoV-2 through breast milk and breastfeeding: a living systematic review. Ann N Y Acad Sci. 2021 Jan;1484(1):32-54.

Florea RM, Sultana CM. COVID-19 and breastfeeding: can SARS-CoV-2 be spread through lactation? Discoveries (Craiova). 2021 Jun 30;9(2):e132. doi: 10.15190/d.2021.11. PMID: 34754901; PMCID: PMC8570917.

Detection of SARS-CoV-2 in Milk From COVID-19 Positive Mothers and Follow-Up of Their Infants

Answer: We appreciate your comment. We hope that the revisions made to the text have highlighted the relevance and contributions of our work compared to previous studies. Specifically, we emphasize the dynamism of the immune composition of breast milk following COVID-19 infection or vaccination, and we incorporate updated findings that integrate the most recent research on infant immune protection.

Submission Date

01 February 2025

Date of this review

19 Feb 2025 23:19:34

Reviewer 3 Report

Comments and Suggestions for Authors

This was an excellent and comprehensive review that contributes greatly to the field as it brings together many concepts related to composition of human milk after maternal exposure to SARS-CoV-2.

I am requesting minor edits.

  1. I recommend slightly change of title such as fallow: Impact of maternal exposure to SARS-CoV-2 on immunological component of breast milk  
  2. The authors analyzed the levels of immunological components of human milk, however skipped the information regarding protein concentration.
  3. Description to Figure 3. I suspect that the first sentence in line 111 should be in line 109. 
  4. Description to Figure 4. I suggest modification of description as fallow: Main techniques for analyzing breast milk
  5. In line 256.  I recommend starting the sentence as follows: Immunoglobulin M (...)
  6. In line 308. I recommend the same number of significant places, namely 7.5% and 3.0%

Author Response

Open Review

Quality of English Language

( ) The English could be improved to more clearly express the research.
(x) The English is fine and does not require any improvement.

Is the work a significant contribution to the field?

Is the work well organized and comprehensively described?

Is the work scientifically sound and not misleading?

Are there appropriate and adequate references to related and previous work?

Is the English used correct and readable?

Comments and Suggestions for Authors

This was an excellent and comprehensive review that contributes greatly to the field as it brings together many concepts related to the composition of human milk after maternal exposure to SARS-CoV-2.

I am requesting minor edits.

  1. I recommend a slight change of title such as the following: Impact of maternal exposure to SARS-CoV-2 on immunological components of breast milk 

Answer: Thanks for the suggestion. We changed the title. 

  1. The authors analyzed the levels of immunological components of human milk; however, however skipped the information regarding protein concentration.

Answer: We appreciate your comment. However, analyzing total protein concentration in breast milk was not the objective of our study; instead, we focused specifically on immunological components related to SARS-CoV-2. Although we considered including studies on lactoferrin due to its immunological relevance, we decided not to address this topic because existing literature presents inconsistencies and conflicts, making it challenging to draw clear conclusions about its impact in this context.

  1. Description to Figure 3. I suspect the first sentence in line 111 should be in line 109. 

Answer: Thank you for your observation. We have corrected it in the text.

  1. Description to Figure 4. I suggest modification of the description as follows: Main techniques for analyzing breast milk

Answer: Thank you for the suggestion. The modification has been made.

  1. In line 256.  I recommend starting the sentence as follows: Immunoglobulin M (...)

Answer: Thank you for the suggestion. The modification has been made.

  1. In line 308. I recommend the same number of significant places, namely 7.5% and 3.0%

Answer:  Thank you for the suggestion. The modification has been made.

Submission Date

01 February 2025

Date of this review

21 Feb 2025 11:19:22

Round 2

Reviewer 1 Report

Comments and Suggestions for Authors

Accept in present form.